



# *ClimLoco1.0*: CLimate variable confidence Interval of Multivariate Linear Observational COnstraint

Valentin Portmann[1, 2, *], Marie Chavent[2], and Didier Swingedouw[1]

[1]Environnements et Paléoenvironnements Océaniques et Continentaux (EPOC) Univ. Bordeaux, CNRS, Pessac, France
[2]Univ. Bordeaux, CNRS, INRIA, Bordeaux INP, IMB, UMR 5251, Talence, France
[*]valentin.portmann@u-bordeaux.fr

**Abstract.** Projections of future climate are key to society's adaptation and mitigation plans in response to climate change. Numerical climate models provide projections, but the large dispersion between them makes future climate very uncertain. To refine it, approaches called observational constraints (*OC*) have been developed. They constrain an ensemble of climate projections by some real-world observations. However, there are many difficulties in dealing with the large literature on *OC*:

the methods are diverse, the mathematical formulation and underlying assumptions used are not always clear, and the methods are often limited to the use of the observation of only one variable. To address these challenges, this article proposes a new statistical model called *ClimLoco1.0*, which stands for "CLimate variable confidence Interval of Multivariate Linear Observational COnstraint". It describes, in a rigorous way, the confidence interval of a projected variable (its best guess associated with an uncertainty at a confidence level) obtained using a multivariate linear *OC*. The article is built up in increasing complexity by expressing in three different cases, the last one being *ClimLoco1.0*, the confidence interval of a projected variable: un-

constrained, constrained by multiple real-world observations assumed to be noiseless, and constrained by multiple real-world observations assumed to be noisy. *ClimLoco1.0* thus accounts for observational noise (instrumental error and climate-internal variability), which is sometimes neglected in the literature but is important as it reduces the impact of the *OC*. Furthermore, *ClimLoco1.0* accounts for uncertainty rigorously by taking into account the quality of the estimators, which depends, for ex-

ample, on the number of climate models considered. In addition to providing an interpretation of the mathematical results, this article provides graphical interpretations based on synthetic data.

## 1 Introduction

Numerical climate models are no exception to the often quoted statement "all models are wrong, but some are useful" from Box (1976). Indeed, their climate *projections* (simulated responses to a scenario of greenhouse gas and aerosol emissions) are

useful to assess future climate change, but they vary widely from one climate model to another (e.g. Bellomo et al., 2021, figure from *IPCC* 2021). There are now several dozen climate models around the world.

To assess the future value of a climate variable, such as global temperature in 2100, a traditional approach is to examine the distribution of projections of the variable simulated by an ensemble of climate models. The climate variable projected by climate models is hereafter referred to as the *projected variable*. The mean and standard deviation, which characterise





the distribution of the projected variable, are usually used to define the so-called *best guess* and *uncertainty* of the projected variable, respectively (Collins et al., 2013). However, this uncertainty is generally high and the best guess may be biased. To incorporate knowledge of real-world observations, statistical methods called observational constraints (*OC*) or emergent constraints (Brunner et al., 2020a; O'Reilly et al., 2024) examine the distribution of the projected variable given real-world observations of an *observable variable* to obtain a *constrained distribution*. Such *OC* approaches are now used in the reports of

the Intergovernmental Panel on Climate Change (*IPCC*) since 2021. They have huge implications for our society. The literature on *OC* methods is flourishing, but there are many difficulties in using them.

Firstly, the large number of existing *OC* methods makes it very difficult to choose one. Some methods average the projections of climate models, with weights that depend on the ability of the models to reproduce real-world observations of a given observable variable (Brunner et al., 2020b; Giorgi and Mearns, 2002; Olson et al., 2018). Some methods use climate models to

learn a relationship between the projected variable and a related observable variable, and use this relationship and a real-world observation of that observable variable to predict the value of the projected variable. This relationship may be linear (Cox et al., 2018; Weijer et al., 2020; Bracegirdle and Stephenson, 2012; Karpechko et al., 2013) or non-linear (Schlund et al., 2020; Li et al., 2021; Forzieri et al., 2021). Other methods statistically give the constrained distribution as the probability density function of the projected variable given the real-world observation of an observable variable (Bowman et al., 2018; Ribes

et al., 2021). This diversity illustrates the lack of consensus on which approach to use. Methods are developed individually and need to be compared to better understand their differences and similarities. However, there are some work of OC methods comparison, for example Brunner et al. (2020a).

Secondly, the approaches and assumptions used to compute the constrained distribution can vary widely between articles and are not always reported. For example, their calculation does not always take into account the instrumental error associated

with the real-world observation. Some papers provide clarification, e.g. Williamson and Sansom (2019), which provided a comprehensive review of the underlying assumptions and uncertainty calculation in *OC* methods based on linear regression. However, some elements are still missing from the literature. For example, the terms "very likely", "unlikely" *etc* used by the *IPCC* (Mastrandrea et al., 2011) come from an underlying statistical model that provides a *confidence interval*, i.e. an interval that contains the projected variable value with a given confidence level. *OC* methods rarely use or describe a confidence

interval. There is therefore a need for a proper statistical description of the theoretical basis of *OC*'s, including confidence intervals, and a full description of the underlying statistical assumptions Hegerl et al. (2021).

Thirdly, *OC* articles often use a *univariate* framework, i.e. they constrain the projected variable using only one observable variable. This may be surprising given the complexity of the climate system, which suggests that the spread between climate model projections may be related to multiple processes. For example, Cox et al. (2018) constrained the equilibrium climate

sensitivity (ECS) using a measure of temperature variability. A few studies, particularly those using non-linear regression, use a *multivariate* framework, but these are still rare. For example, Schlund et al. (2020) constrained future spatio-temporal *GPP* (Gross Primary Production) by past spatio-temporal *GPP* and temperature.

To address these challenges, this article proposes a statistical model called *ClimLoco1.0*, which stands for "CLimate variable confidence Interval of Multivariate Linear Observational COnstraint". *ClimLoco1.0* expresses the confidence interval of a





climate variable constrained using a linear multivariate observational constraint that takes into account observational noise. *ClimLoco1.0* can also be used in univariate instead of multivariate. This is the first version, 1.0, calling for further improvements to better account for all uncertainties. This article builds *ClimLoco1.0* progressively by increasing complexity by expressing the confidence interval of the projected variable unconstrained (Sect. 2), constrained by noiseless real-world observations (Sect. 3), and finally constrained by a noisy real-world observation (Sect. 4). The latter corresponds to *ClimLoco1.0*. Since the

devil can be in the details, the article presents the statistical procedure in a rigorous and clear manner, based on mathematical demonstrations. Moreover, the use of this complex statistical procedure is justified by illustrations of the underestimation of the uncertainty usually made in the literature by not using rigorous *CI*'s. These results are then compared with some of the most widely used methods in the literature (Sect. 5): statistical methods as in Bowman et al. (2018) or in Ribes et al. (2021), and methods based on linear regression as in Cox et al. (2018).

## 70    2   Confidence interval of $Y$ unconstrained

In order to anticipate society's adaptation and mitigation plans in response to climate change, it may be necessary to estimate the value of a future variable called *projected variable* and denoted $Y$, e.g. the global temperature in 2100. A common approach is to use an ensemble of climate model projections, e.g. *CMIP6* (Coupled Model Intercomparison project version 6), which give different values of $Y$. $Y$ is therefore a random variable: the dispersion between the climate model projections comes from

the randomness of $Y$.

To properly estimate the value of $Y$, this section defines the *confidence interval* (*CI*) of $Y$. It provides a best guess of $Y$ value (centre of the interval) and an associated uncertainty (width of the interval) at a given confidence level. This section gradually builds up the *CI* of $Y$ in increasing complexity. Firstly, it defines the *probability interval* (*PI*) of $Y$ obtained assuming that the theoretical distribution of $Y$ is known. Secondly, it defines the *CI* of $Y$ obtained using this distribution estimated on an

ensemble of climate models. These two types of intervals are illustrated and interpreted using a synthetic example.

As stated above, the *PI* of $Y$ is build using the theoretical distribution of $Y$. Here, this distribution is assumed to be Gaussian: $Y \sim \mathcal{N}(\mu_Y, \sigma_Y^2)$, where $\mu_Y$ and $\sigma_Y^2$ are respectively the expectation and variance of $Y$. The *PI* of $Y$ is the interval that contains $Y$ values with a probability of $1 - \alpha$:

$$\mathbb{P}(\mu_Y - z\,\sigma_Y \leq Y \leq \mu_Y + z\,\sigma_Y) = 1 - \alpha, \tag{1}$$

where $z$ is the quantile of order $1 - \alpha/2$ of a distribution $\mathcal{N}(0,1)$. For example, the 90% *PI* ($\alpha = 0.1$) is obtained with $z = 1.65$. In the *IPCC*, this 90% probability corresponds to the term "very likely", while "likely" stands for the 66% probability, *etc*. In the following, the *PI* of $Y$ associated with a probability of $1 - \alpha$ described in Eq. (1) is denoted as:

$$PI_{1-\alpha}(Y) = [\mu_Y \pm z\,\sigma_Y]. \tag{2}$$

In fact, the expectation $\mu_Y$ and the standard deviation $\sigma_Y$ are unknown. The *PI* described by Eq. (2) is therefore unknown.

However, $\mu_Y$ and $\sigma_Y$ can be estimated from an ensemble of climate model projections, for example from *CMIP6*. This ensemble of $M$ climate model projections yields a sample of $M$ random variables, denoted $(Y_1, ..., Y_M)$. These random variables are





assumed to be independent and to follow the same law as $Y$, which is assumed to be $\mathcal{N}(\mu_Y, \sigma_Y^2)$. The classical estimators of the expectation $\mu_Y$ and variance $\sigma_Y^2$ are:

$$\hat{\mu}_Y = \frac{1}{M}\sum_{i=1}^{M} Y_i, \tag{3}$$

$$\hat{\sigma}_Y^2 = \frac{1}{M-1}\sum_{i=1}^{M}(Y_i - \hat{\mu}_Y)^2. \tag{4}$$

The literature usually replaces $\mu_Y$ and $\sigma_Y$ by their estimators $\hat{\mu}_Y$ and $\hat{\sigma}_Y$ to estimate the *PI* $[\mu_Y \pm z\,\sigma_Y]$, which gives the interval $[\hat{\mu}_Y \pm z\,\hat{\sigma}_Y]$. This interval has no clear statistical meaning. In fact, $\hat{\mu}_Y$ and $\hat{\sigma}_Y$ are random variables that depend on $M$, the number of climate models used. The quality of these two estimators affects the quality of the interval. It can be shown (see appendix B) that using these estimators $\hat{\mu}_Y$ and $\hat{\sigma}_Y$, the values of $Y$ are contained in the following interval with a probability

of $1 - \alpha$:

$$\mathbb{P}\left(\hat{\mu}_Y - t^{M-1}\hat{\sigma}_Y\sqrt{1 + \frac{1}{M}} \le Y \le \hat{\mu}_Y + t^{M-1}\hat{\sigma}_Y\sqrt{1 + \frac{1}{M}}\right) = 1 - \alpha, \tag{5}$$

where $t^{M-1}$ is the quantile of a Student distribution with $M - 1$ degrees of freedom associated with the probability $1 - \alpha$. For example, with a confidence level of 90% ($\alpha = 0.1$), $t^{30} = 1.70$ and $t^5 = 2.02$.

A subtle point is that this interval described in Eq. (5) is not a *probability interval* (*PI*) but a so called *confidence interval*
(*CI*). For example, the 90% *PI* of $Y$ is an interval that has a 90% probability of containing $Y$ values. It has *deterministic* bounds which frame a random variable. While the CI of $Y$ has *random* bounds, which also frame a random variable. In fact, the *CI* of $Y$ described in Eq. (5) has random bounds because $\hat{\mu}_Y$ and $\hat{\sigma}_Y$ are random variables. Thus, different sample realisations, e.g. from different ensembles of climate models, will lead to different realisations of this *CI*. Instead, there is a *confidence* of 90% that one realisation of the *CI* contains $Y$. In other words, out of 100 realisations of the 90% *CI*, 90 should contain the
value of the random variable $Y$. This is illustrated in Fig. 1, which shows 100 realisations of this CI of $Y$ (error bars) at a 90% confidence level, as well as 100 realisations of $Y$ (red dots). This specific type of *CI* is also often called a *prediction interval*.

The *CI* of $Y$ associated with a confidence level of $1 - \alpha$ is then denoted:

$$CI_{1-\alpha}(Y) = \left[\hat{\mu}_Y \pm t^{M-1}\hat{\sigma}_Y\sqrt{1 + \frac{1}{M}}\right]. \tag{6}$$

Now that the *CI* of $Y$ is defined, it let us study the effect on it of the number of climate models considered, $M$. Throughout
this paper, the same synthetic example is used, defined in appendix C. It provides the theoretical *PI* of $Y$, which is unknown in reality and estimated by the *CI* of $Y$. It also provides one realisation of the *CI* of $Y$ using a small ($M = 5$) sample $(Y_1, ..., Y_M)$, and another using a large sample ($M = 30$). These different samples can, for example, represent different ensembles of climate models (*CMIP5*, *CMIP6*, *HighResMIP*). Figure 2 shows the probability interval of $Y$, $PI_{1-\alpha}(Y)$ defined in Eq. (2), and the realisations of the two *CI* of $Y$ ($M = 5$ and $M = 30$), $CI_{1-\alpha}(Y)$ defined in Eq. (6). In reality the *PI* is unknown. This synthetic
example allows us to compare the estimates with the truth.





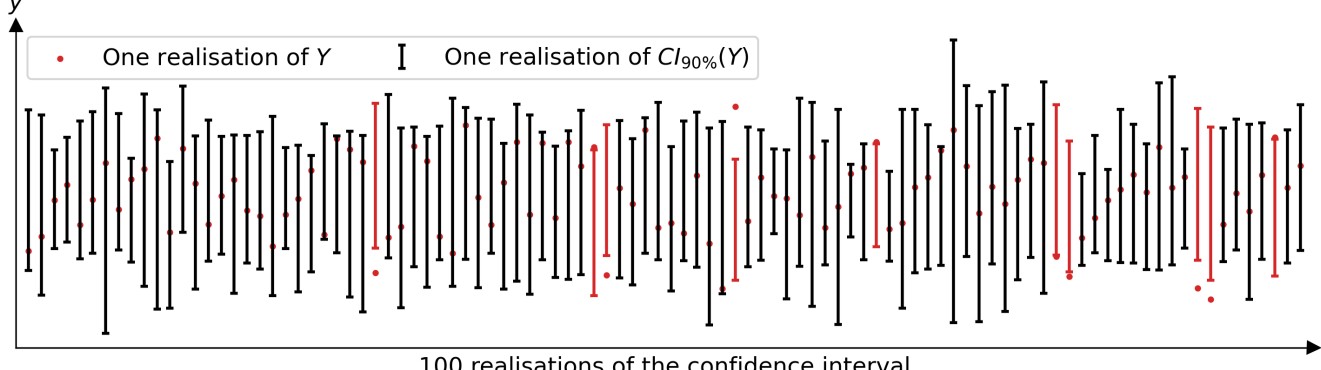

**Figure 1.** 100 random realisations of the 90% confidence interval (*CI*) of $Y$, $CI_{90\%}(Y)$ described in Eq. (6), and 100 realisations of the random variable $Y$ (red dots). Each realisation of the *CI* comes from a sample of $M = 10$ random realisations of $Y$. Since the confidence level is 90%, it is expected that 90 out of 100 *CI*s realisations contain the realisation of $Y$, which is the case in this figure. The 10 *CI*s that did not contain the realisation of $Y$ are shown in red.

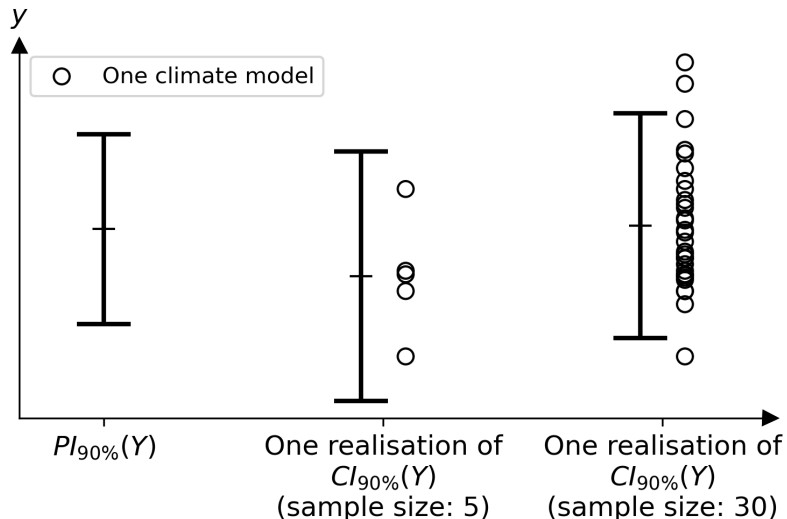

**Figure 2.** Synthetic example comparing the 90% probability interval (*PI*) of $Y$ (left), described in Eq. (2), with two realisations of the 90% confidence interval (*CI*) of $Y$ (middle and right), described in Eq. (6). The first realisation is obtained with the small sample of $M = 5$ climate models, while the second with the large sample of $M = 30$. The *PI* is unknown in reality, it is the truth to compare with. The details of the data simulation are given in appendix C.

There are two important remarks about the *CI* of $Y$ described in Eq. (6). Firstly, it converges in probability to the *PI* of $Y$ described in Eq. (2) as $M$, the number of climate models considered, increases. Indeed, as $M$ becomes large, the estimates



$\hat{\mu}_Y$ and $\hat{\sigma}_Y$ (Eq. (3) and Eq. (4)) converge (in probability) to $\mu_Y$ and $\sigma_Y$, and the Student quantile converges to a Gaussian quantile. This is illustrated in Fig. 2 by comparing the *PI* of $Y$ (left error bar) with the realisations of the *CI* of $Y$ (middle and

right error bars). Indeed, the large sample gives a *CI* of $Y$ (right interval, at $[0.1 \pm 1.9]$) closer to the *PI* of $Y$ (left interval, at $[0 \pm 1.6]$) than the small sample (middle interval, at $[-0.8 \pm 2.2]$). To accurately estimate both the centre and the width of the *CI* of $Y$, which represent the best guess and the uncertainty respectively, it is therefore necessary to have as large a sample as possible. Secondly, the fewer the models, the larger the *CI* of $Y$. It is intuitive that estimating the *CI* of $Y$ with less data will give a more uncertain result. Indeed, in Eq. (6), the terms $t^{M-1}$ and $\sqrt{1 + \frac{1}{M}}$ are larger when $M$ is smaller. These two

aspects highlight the importance of having as many climate models as possible. However, the climate models considered must be independent and the simulated variables must follow the same distribution as the real variables, two assumptions necessary for the calculation that are not fully satisfied (Knutti et al., 2017).

In the literature, the *PI* of $Y$, i.e. $[\mu_Y \pm z\,\sigma_Y]$, is often estimated as the empirical interval $[\hat{\mu}_Y \pm z\,\hat{\sigma}_Y]$. However, as seen previously, this interval has no statistical basis, whereas $CI_{1-\alpha}(Y) = \left[\hat{\mu}_Y \pm t^{M-1}\,\hat{\sigma}_Y\,\sqrt{1 + \frac{1}{M}}\right]$, contains $Y$ value with a given

probability. The relative error of the interval width caused by using the wrong interval $[\hat{\mu}_Y \pm z\,\hat{\sigma}_Y]$ instead of the $CI$ is therefore quantified as relative error in the width of this wrong interval ($z\,\sigma_Y$) compared to the width of the $CI$ ($t^{M-1}\,\hat{\sigma}_Y\,\sqrt{1 + \frac{1}{M}}$):

$$E_1 = \frac{\left| z - t^{M-1}\,\sqrt{1 + \frac{1}{M}} \right|}{t^{M-1}\,\sqrt{1 + \frac{1}{M}}}. \tag{7}$$

This relative error, which depends on the sample size ($M$) and the confidence level ($1 - \alpha$) controlling $z$ and $t$, is plotted as a function of these two parameters in Fig. 3. For typical sample sizes of ensembles of climate models, between 5 and 50, the

relative error is between 3% and 30%. For example, with a confidence level of 68% ($z = 1$, "likely" in *IPCC* language) and a sample size of 20 climate models, the relative error is 5%. Since the width of the interval represents the uncertainty, this means that the uncertainty is underestimated by 5%, which is even higher for smaller sample sizes or higher confidence levels. This highlights the importance of using the rigorous formula provided in this article to express uncertainty more accurately.

Without even mentioning observational constraints, this section provides statistically sound formulae for estimating an inter-

val that contains the value of a future variable from the projections of an ensemble of $M$ climate models at a given confidence level, using confidence intervals. This brings a rigourous lighting to climate science, where simple mean and standard deviation are commonly used. The next part applies the same methodology to a linear observational constraint.

## 3 Confidence interval of $Y$ constrained by a noiseless observation

Observational constraint (*OC*) methods have been developed to estimate more accurately the value of a projected variable $Y$.

These methods "constrains" the distribution of $Y$ by a "real-world" observation, denoted $x_0$, of an observable variable $X$. In this section, the real-world observation is assumed to be noiseless (no observational noise, e.g. due to instrumental errors). This assumption is relaxed in the next section which defines *ClimLoco1.0*. The general formulation presented in this article can be





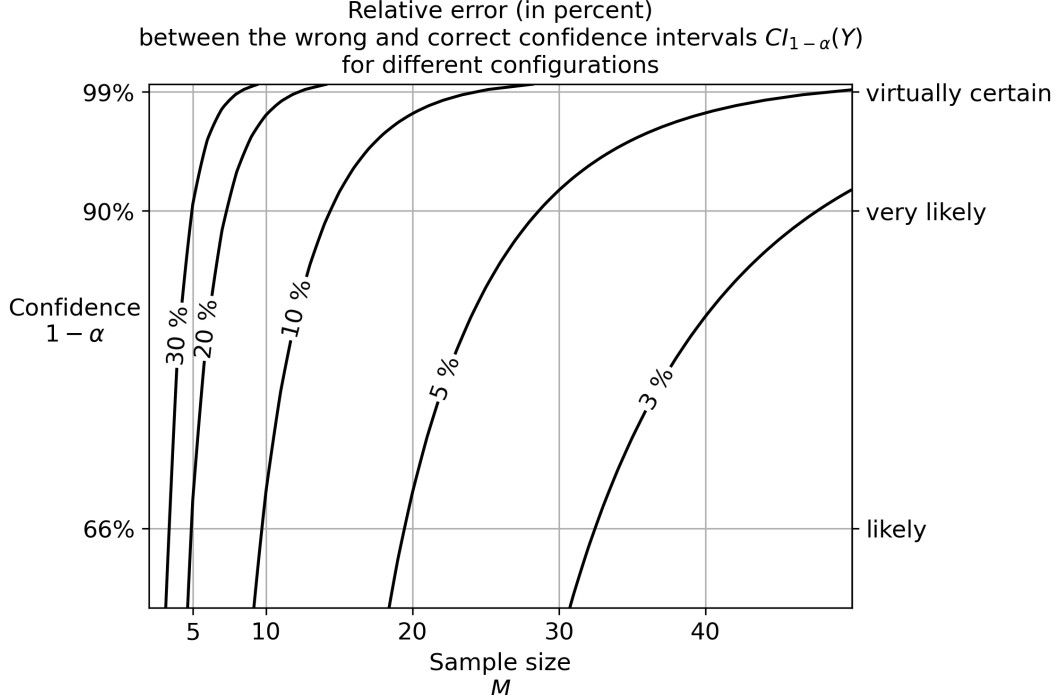

**Figure 3.** Uncertainty quantification errors caused by using the wrong interval instead of the correct one, or $[\hat{\mu}_Y \pm z\,\hat{\sigma}_Y]$ instead of $\left[\hat{\mu}_Y \pm t^{M-1}\hat{\sigma}_Y\sqrt{1+\frac{1}{M}}\right]$. This relative error is described in Eq. 7. The contours correspond to relative errors of 30%, 20%, 10%, 5% and 3%.

applied to the choice of any arbitrary variables $X$ and $Y$. The variable $Y$ constrained by the observation $x_0$ of $X$ is written as $Y|X = x_0$.

This section gradually builds up the *CI* of $Y|X = x_0$ in increasing complexity. Firstly, it defines the probability interval (*PI*) of $Y$ obtained using the theoretical distribution of $Y|X = x_0$ by assuming that this distribution is known. Secondly, it defines the *CI* of $Y|X = x_0$ obtained using this distribution estimated on an ensemble of climate models. These two types of intervals are illustrated and interpreted using a synthetic example.

     As stated above, the *PI* of $Y|X = x_0$ is build using the theoretical distribution of $Y|X = x_0$. Here, this distribution is

assumed to be Gaussian: $Y|X = x_0 \sim \mathcal{N}(\mu_{Y|X=x_0}, \sigma^2_{Y|X=x_0})$, where $\mu_{Y|X=x_0}$ and $\sigma^2_{Y|X=x_0}$ are respectively the expectation and variance of $Y|X = x_0$. In the following, the *PI* of $Y|X = x_0$ associated with a probability of $1 - \alpha$ is denoted as:

$$PI_{1-\alpha}(Y|X = x_0) = \left[\mu_{Y|X=x_0} \pm z\,\sigma_{Y|X=x_0}\right], \qquad (8)$$

where $z$ is the quantile of order $1 - \alpha/2$ of a distribution $\mathcal{N}(0,1)$. As mentioned in the introduction, many articles in the literature use univariate *OC*, i.e. only one observable variable $X$ is used to constrain $Y$. This can be very limiting, especially

when $Y$ depends on many processes, which is often the case in climate science. Therefore, an important contribution of this





article is to give the formulation in a multivariate form, i.e. where $Y$ is constrained by several observable variables at the same time. However, for the sake of clarity, only the results for the univariate formalisations are presented in the main part of the article. The multivariate formalisations are given in appendix table A1. In order to express the terms $\mu_{Y|X=x_0}$ and $\sigma^2_{Y|X=x_0}$ in Eq. (8), it is used a linear regression framework:

$$Y = \mathbb{E}[Y|X] + \varepsilon, \tag{9}$$


$$\text{where } \mathbb{E}[Y|X] = a_0 + a_1 X, \varepsilon \sim \mathcal{N}(0, \sigma_\varepsilon),$$

and where the coefficients $a_0$ and $a_1$ are the intercept and the slope of the linear regression of $Y$ on $X$, respectively, and $\varepsilon$ is a random variable representing the regression error with $\sigma_\varepsilon$ its standard deviation. Using this linear regression model, it can be shown (see the proof in appendix D) that the terms $\mu_{Y|X=x_0}$ and $\sigma^2_{Y|X=x_0}$ can be expressed as:

$$\mu_{Y|X=x_0} = a_0 + a_1 x_0 \tag{10}$$


$$= \mu_Y + \rho \frac{\sigma_Y}{\sigma_X}(x_0 - \mu_X), \tag{11}$$

$$\sigma^2_{Y|X=x_0} = \sigma^2_\varepsilon \tag{12}$$

$$= (1 - \rho^2)\sigma^2_Y, \tag{13}$$

where $\mu_X$ and $\mu_Y$ are the expectations of $X$ and $Y$, $\sigma_X$ and $\sigma_Y$ are the standard deviations of $X$ and $Y$, and $\rho$ is the linear correlation between $X$ and $Y$. The *PI* of $Y$ constrained by $X = x_0$ described by Eq. (8) can then be rewritten:


$$PI_{1-\alpha}(Y|X = x_0) = [a_0 + a_1 x_0 \pm z \sigma_\varepsilon]. \tag{14}$$

To illustrate this, it is use the same synthetic case study as before, detailed in appendix C. The *PI* of $Y$ constrained is shown in Fig. 4.a and 4.b (in red), and is compared with the *PI* of $Y$ unconstrained (in black) in Fig. 4.b. The constraint on $Y$ has two effects: (a) it changes the best guess (centre of the interval), and (b) it reduces the uncertainty (width of the interval).

(a) When $Y$ is constrained ($PI(Y|X = x_0)$), it has a different best guess (centre of interval) than when it is unconstrained

($PI(Y)$). We interpret this in two different ways, using Eq. (10) and Eq. (11). The first equation gives a graphical interpretation: the constrained expectation of $Y$ is directly the real-world observation fed into the regression. This is illustrated in Fig. 4.a. The second equation is useful to understand the correction between the best guess of $Y$ constrained and unconstrained: $\mu_{Y|X=x_0} - \mu_Y = \rho \frac{\sigma_Y}{\sigma_X}(x_0 - \mu_X)$. It depends on two terms: the regression slope $\rho \frac{\sigma_Y}{\sigma_X}$ which depends in particular on the correlation between $X$ and $Y$, and the difference between the real-world observation and the theoretical centre of the climate models distribution on

$X$. It is called here $(x_0 - \mu_X)$ the *theoretical multi-model bias*. In other words, the constrained best guess of $Y$ ($\mu_{Y|X=x_0}$) is a corrected version on the unconstrained best guess of $Y$ ($\mu_Y$), knowing the theoretical multi-model bias of $X$ ($x_0 - \mu_X$) and the relationship between $X$ and $Y$ ($\rho \frac{\sigma_Y}{\sigma_X}$). In the example of Fig. 4.a, there is a positive theoretical multi-model bias associated with a positive relationship between $X$ and $Y$, thus a correction to a higher best guess ($\mu_{Y|X=x_0} > \mu_Y$). Observational constraints are generally used to reduce uncertainty, but the correction of the best guess between constrained and unconstrained is very

important and should not be forgotten, as it allows to correct the multi-model bias.





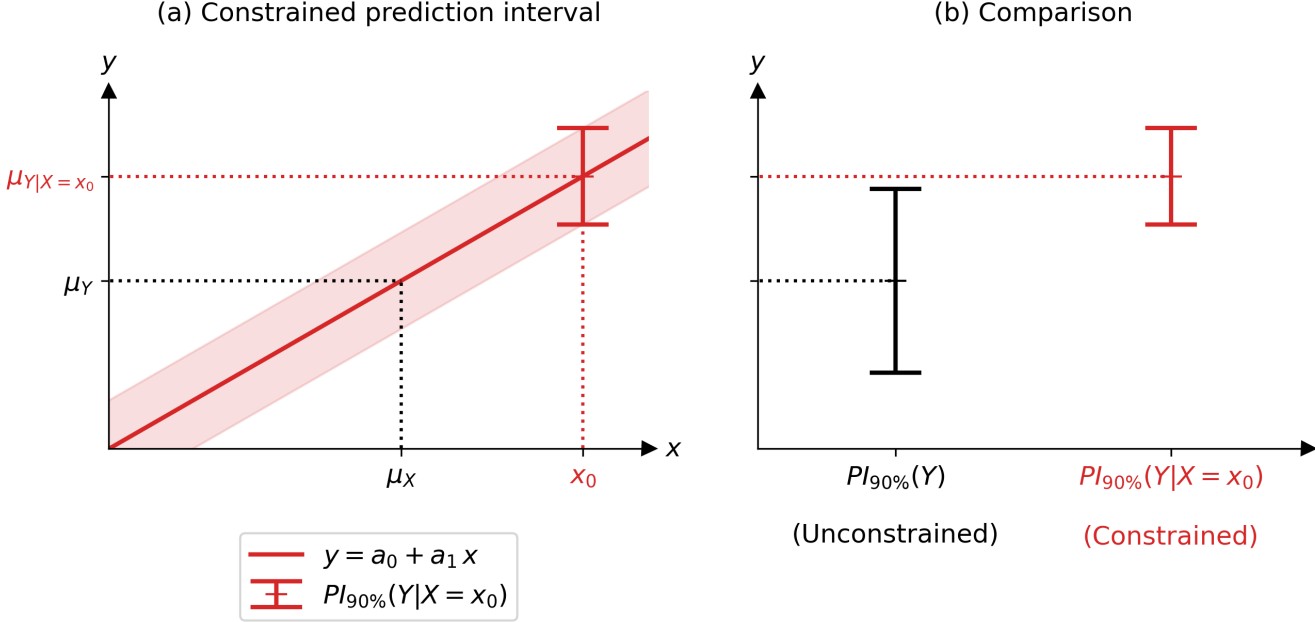

**Figure 4.** (a) Example showing the 90% probability interval (*PI*) of the projected variable $Y$ constrained by the observation $x_0$ of an observable variable $X$, as described by Eq. (14). (b) Comparison between the 90% *PI* of $Y$ constrained (red) and unconstrained (black) as described by Eq. (14) and Eq. (2) respectively. The values of means, variances *etc* are given in appendix C.

(b) When $Y$ is constrained, it has a lower uncertainty (width of $PI(Y|X=x_0)$) than when it is unconstrained (width of $PI(Y)$). We interpret this in two different ways, using Eq. (12) and Eq. (13). The first equation provides a graphical interpretation: the uncertainty of $Y$ constrained is directly the regression error. The 90% regression error is represented by the red tube Fig. 4.a. The second equation expresses the variance of $Y$ ($Y|X=x_0$) constrained as the variance of $Y$ unconstrained

multiplied by $1-\rho^2$, which is between $0$ and $1$. The uncertainty of $Y$ constrained is therefore smaller than the uncertainty of $Y$ unconstrained: this is the desired reduction in uncertainty. The stronger the correlation between $X$ and $Y$, the greater the reduction in uncertainty. In the example shown in Fig. 4.b, the strong correlation (0.85) between $X$ and $Y$ reduces the uncertainty well, the red interval is narrower than the black one.

The use of the *PI* of $Y$ constrained, $PI(Y|X=x_0)$ (described), requires knowledge of the theoretical parameters $a_0$,

$a_1$ and $\sigma_\varepsilon$, which are unknown in reality. To estimate them, it is used an ensemble of $M$ climate models. It is written $(X_1, Y_1), ..., (X_M, Y_M)$ as a sample of $M$ pairs of random variables $(X, Y)$. They are assumed to be independent and to follow the same law as $(X, Y)$, which is assumed to be bivariate Gaussian. This sample allows to define the estimators $\hat{a}_0$, $\hat{a}_1$ and $\hat{\sigma}_\varepsilon$ of $a_0$, $a_1$ and $\sigma_\varepsilon$ (see formulas in appendix table A2). To estimate $PI(Y|X=x_0)$, one may want to replace the theoretical parameters by the estimated ones, which gives the following interval: $[\hat{a}_0 + \hat{a}_1 x_0 \pm z \hat{\sigma}_\varepsilon]$. However, as seen in the





previous section, this interval has no statistical meaning. Instead, it is shown in appendix E that the estimated parameters lead to the following confidence interval (*CI*) of $Y|X = x_0$:

$$CI_{1-\alpha}(Y|X = x_0) = \left[ \hat{a}_0 + \hat{a}_1 x_0 \pm t^{M-2} \hat{\sigma}_\varepsilon \sqrt{1 + \frac{1}{M} + \frac{(x_0 - \hat{\mu}_X)^2}{M \hat{\sigma}_X^2}} \right]. \tag{15}$$

The corresponding expression when $X$ is multivariate is given in appendix table A1.

To illustrate these mathematical results, Fig. 5 uses the same synthetic case study as before. It shows the realisations of two

samples $(X_1, Y_1), ..., (X_M, Y_M)$, one of size $M = 5$, and the other of size $M = 30$. The realisation of each sample corresponds to each row. As shown in Fig. 5.a, these two different sample realisations lead to two different realisations of the estimated linear relationship $y = \hat{a}_0 + \hat{a}_1 x$ (red line) and the constrained confidence intervals (red interval). The red shading represents the $CI_{90\%}(Y|X = x_0)$ obtained for different positions of the observation $x_0$. Figure 5.b compares the realisations of the *CI* of $Y$ unconstrained $CI_{90\%}(Y)$ and constrained $CI_{90\%}(Y|X = x_0)$, and the *PI* of $Y$ constrained $PI_{90\%}(Y|X = x_0)$.

On the one hand, there are two similarities between the *CI* of $Y$ constrained $(Y|X = x_0)$ and unconstrained $(Y)$. Firstly, the *CI* of $Y$ constrained described by Eq. (15) converges (in probability) to the *PI* of $Y$ constrained described by Eq. (8) as the sample size $M$ increases. Consequently, the *CI* obtained from a large sample (second pannel) is closer to the *PI* than the one obtained from a small sample (first pannel), as shown in Fig. 5.b. Secondly, the CI of $Y|X = x_0$ is larger when the sample size $M$ is smaller, due to the term $t^{M-2} \sqrt{1 + \frac{1}{M} + \frac{(x_0 - \hat{\mu}_X)^2}{M \hat{\sigma}_X^2}}$ in Eq. (15). To summarise these two similarities between the

unconstrained and constrained cases, a larger sample leads to a more correct and precise estimate of $Y$.

On the other hand, there are two important differences between the *CI* of $Y$ constrained and unconstrained. Firstly, the centre of $CI(Y|X = x_0)$ is the observation fed into the regression, as described in Eq. (15). Using the previous equations, the difference between the centre (best guess) of the *CI* of $Y$ constrained and unconstrained can be expressed as $\hat{\mu}_{Y|X=x_0} - \hat{\mu}_Y = \hat{a}_1 (x_0 - \hat{\mu}_X)$. This correction of the best guess depends on the estimated slope between $X$ and $Y$, $\hat{a}_1$, and on what is called

here the multi-model mean bias at $X$, $(x_0 - \hat{\mu}_X)$. In other words, the constraint corrects the multi-model bias on $Y$, knowing the multi-model bias on $X$ and the relationship between $X$ and $Y$. This is illustrated in Fig. 5. Secondly, there is a difference in the square root term between the *CI* of $Y$ constrained and unconstrained. The *CI* of $Y$ constrained is larger by the amount $(x_0 - \hat{\mu}_X)^2 / M \hat{\sigma}_X^2$. If the observation is far from the samples, this quantity is large, which makes the interval width (uncertainty) larger. In other words, the linear relationship is more uncertain away from the samples, in unknown territory. Furthermore, the

latter term is multiplied by $1/M$, which means that the linear relationship is more certain when obtained from a large sample size. This is illustrated in Fig. 5.a on the small sample (first row): the constrained confidence interval grows rapidly as one move away from the samples (back circles).

In summary, the equations and figures show the two benefits sought in *OC*: there is a correction to the best guess and a reduction in uncertainty, between the *CI* of $Y$ unconstrained and constrained. To maximise the reduction in uncertainty, the is

a need for as many (independent) climate models as possible.



**Figure 5.** Synthetic example showing, (a) the first column, two realisations of the 90% confidence interval (*CI*) of $Y$ constrained by the observation $x_0$ of $X$, described in Eq. (15). The two realisations come from two different samples $(X_1, Y_1), ..., (X_M, Y_M)$ of size $M = 5$ and $M = 30$ (black circles) and correspond to the two rows of the figure. The estimated linear regression and its 90% error are shown as the red line and shade, respectively. (b), the second column, compares in red the confidence (middle) and probability (right) intervals of the constrained $Y$. The larger sample (second row) gives a better *CI* than the small one (first row), which is closer to *PI*. (b) also compares the *CI* of $Y$ constrained (middle) and unconstrained (left, in black). The details of the data simulation are given in appendix C. $\hat{\mu}_X$ and $\hat{\mu}_Y$ are the unconstrained means of $X$ and $Y$, while $\hat{\mu}_{Y|X=x_0}$ is the constrained mean of $Y$. The observation $x_0$ is assumed to be noiseless.

As seen previously, to get a real estimate of the *PI* of $Y$ constrained, namely $[a_0 + a_1 x_0 \pm z \sigma_\varepsilon]$, the correct approach is to use the *CI* of $Y$ constrained, namely $\left[ \hat{a}_0 + \hat{a}_1 x_0 \pm t^{M-2} \hat{\sigma}_\varepsilon \sqrt{1 + \frac{1}{M} + \frac{(x_0 - \hat{\mu}_X)^2}{M \hat{\sigma}_X^2}} \right]$. However, the literature sometimes uses $[\hat{a}_0 + \hat{a}_1 x_0 \pm z \hat{\sigma}_\varepsilon]$, which has no statistical basis. The relative error in the interval width caused by using the wrong one instead of the correct one is therefore quantified as:





$$E_2 = \frac{\left| z - t^{M-2} \sqrt{1 + \frac{1}{M} + \frac{(x_0 - \hat{\mu}_X)^2}{M \hat{\sigma}_X^2}} \right|}{t^{M-2} \sqrt{1 + \frac{1}{M} + \frac{(x_0 - \hat{\mu}_X)^2}{M \hat{\sigma}_X^2}}}.$$ (16)

This relative error described by Eq. (16) depends on three parameters: (i) the sample size, $M$, (ii) the confidence level, $1 - \alpha$, which controls $z$ and $t$, and (iii) the standardised real-world observation, $\frac{(x_0 - \hat{\mu}_X)^2}{\hat{\sigma}_X^2}$. The relative error is shown in Fig. 6 for a fixed confidence level of 68%, as a function of $M$ (x-axis) and $\frac{(x_0 - \hat{\mu}_X)^2}{\hat{\sigma}_X^2}$ (y-axis). With a typical sample size of climate model ensembles between 5 and 50, the relative error is between 3% and 30%. In other words, using the wrong interval instead

of the correct one implies an underestimation of the uncertainty between 3% and 30%. For example, using an ensemble of $M = 20$ climate models, the error starts at 5% and can easily exceed 10% if the observation is far from the ensemble of climate models (y-axis). This highlights the need to rigorously consider the performance of the estimators in order to correctly estimate the uncertainty using the rigorous *CI*.

## 4   *ClimLoco1.0*

The previous results were obtained under the assumption that the real-world observation $x_0$ is not noisy. In reality, $x_0$ is affected by observational noise, which is taken into account in this section, inspired by the theory of measurement error models (Fuller, 2009). Some papers define observational noise as internal variability (e.g. Brunner et al., 2020b), others as measurement error (e.g. Hall et al., 2019), and others as both (e.g. Ribes et al., 2021). We argue here that both internal variability and measurement error should be taken into account, as both affect the real-world observation. Let $X^N$ be the noisy version of $X$, linked by the

noise model defined in Bowman et al. (2018):

$$X^N = X + N, \text{ with } N \sim \mathcal{N}(0, \sigma_N^2) \text{ and } N \perp\!\!\!\perp X,$$ (17)

where $N$ is a random variable representing the observation noise, assumed to be Gaussian, centred and independent of $X$. Its variance $\sigma_N^2$ is assumed to be known. The projected variable $Y$ constrained by the observation $x_0^N$ of $X^N$ affected by the observation noise is denoted $Y | X^N = x_0^N$.

This section constructs *ClimLoco1.0*, which is the confidence interval of $Y | X^N = x_0^N$, in increasing complexity, following the same steps as in the previous two sections. Firstly, it defines the probability interval (*PI*) of $Y | X^N = x_0^N$ obtained using the theoretical distribution of $Y | X^N = x_0^N$ by assuming that the distribution of $Y | X^N = x_0^N$ is known. Secondly, it defines the *CI* of $Y | X^N = x_0^N$ obtained using this distribution estimated on an ensemble of climate models. These two types of intervals are illustrated and interpreted using a synthetic example. These different steps that construct *ClimLoco1.0* are crucial to define

and understand with rigor the best guess and uncertainty of any variable constrained by a noisy observation.





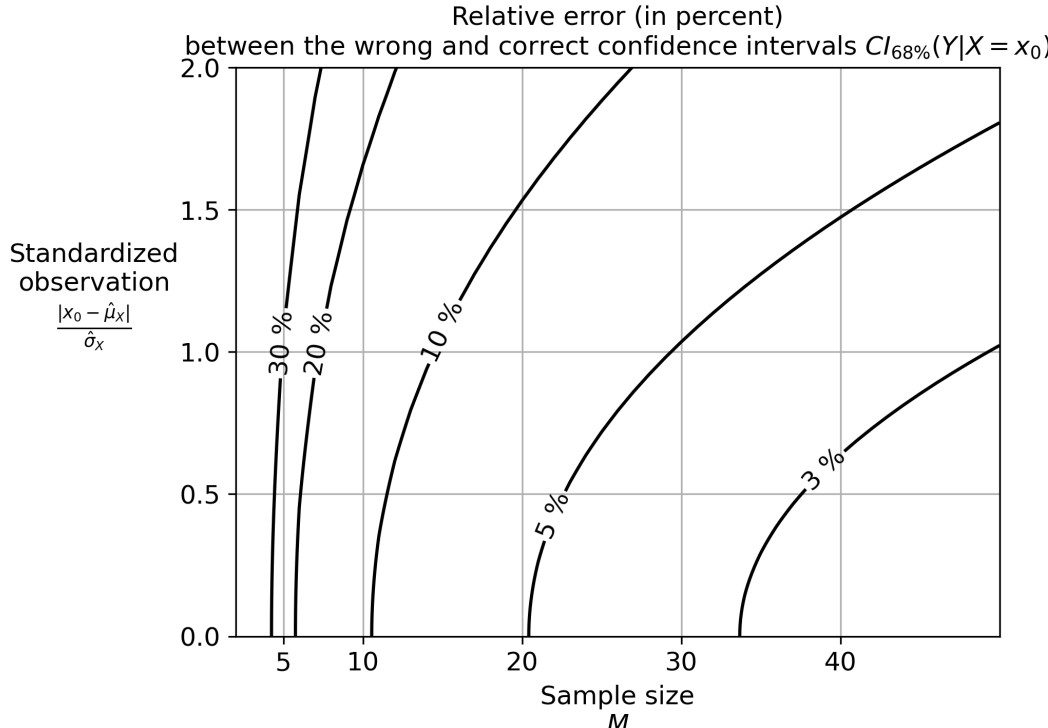

**Figure 6.** Uncertainty quantification error when constraining $Y$ made by using the wrong interval instead of the correct one, i.e. $[\hat{a}_0 + \hat{a}_1 x_0 \pm z\,\hat{\sigma}_\varepsilon]$ instead of $\left[\hat{a}_0 + \hat{a}_1^T x_0 \pm t^{M-2}\,\hat{\sigma}_\varepsilon \sqrt{1 + \frac{1}{M} + \frac{(x_0 - \hat{\mu}_X)^2}{M\,\hat{\sigma}_X^2}}\right]$. This error is described in Eq. (16). The error values are shown by the contours between 3 and 30%. They are given as a function of the sample size (x-axis) and the distance between the observation and the multi-model ensemble. The confidence level is fixed at $68\%$ (i.e. $z = 1$), a value often used in the literature.

As stated above, the *PI* of $Y|X^N = x_0^N$ is build using the theoretical distribution of $Y|X^N = x_0^N$. Here, this distribution is assumed to be Gaussian: $Y|X^N = x_0^N \sim \mathcal{N}(\mu_{Y|X^N=x_0^N}, \sigma^2_{Y|X^N=x_0^N})$, where $\mu_{Y|X^N=x_0^N}$ and $\sigma^2_{Y|X^N=x_0^N}$ are respectively the expectation and variance of $Y|X^N = x_0^N$. The following interval is the *PI* of $Y|X^N = x_0^N$, i.e. it contains $Y|X^N = x_0^N$ values with a probability of $1 - \alpha$:

$$PI_{1-\alpha}(Y|X^N = x_0^N) = \left[\mu_{Y|X^N=x_0^N} \pm z\,\sigma_{Y|X^N=x_0^N}\right], \qquad (18)$$

where $z$ is the quantile of order $1 - \alpha/2$ of a distribution $\mathcal{N}(0,1)$. This interval contains realisations of $Y$ with a given confidence $1 - \alpha$ controlling $z$. To express the parameters $\mu_{Y|X^N=x_0^N}$ and $\sigma^2_{Y|X^N=x_0^N}$, it is used a linear regression framework as in the previous section:

$$Y = \mathbb{E}[Y|X^N] + \varepsilon^N,$$

where $\mathbb{E}[Y|X^N] = b_0 + b_1 X^N$, $\varepsilon^N \sim \mathcal{N}(0, \sigma_{\varepsilon^N})$,

$$(19)$$



and where $b_0$ and $b_1$ are the intercept and slope of the linear model, and $\varepsilon^N$ is a random variable representing the regression error with $\sigma_{\varepsilon^N}$ its standard deviation. This linear regression is the regression of $Y$ on $X^N$. But climate models do not suffer from observational noise (instrumental error and internal variability): they provide realisations of $X$, not $X^N$. In fact, climate models do not suffer from instrumental error, but they can be affected by internal variability. However, the impact of internal variability can be reduced for example by averaging the members of this climate model (different realisations run from different

initialisations). As climate models provide realisations of $X$ and not $X^N$, it can be difficult to express the linear coefficients $b_0$ and $b_1$. However, using the model noise described by Eq. (17), which relates $X^N$ to $X$, it is possible to obtain the expressions of $b_0$ and $b_1$ and hence the expression of $\mu_{Y|X^N=x_0^N}$ and $\sigma^2_{Y|X^N=x_0^N}$ . In fact, it can be shown (see Appendix F) that:

$$\mu_{Y|X^N=x_0^N} = b_0 + b_1\,x_0^N \tag{20}$$

$$= \mu_Y + \rho \frac{\sigma_Y}{\sigma_X} \frac{1}{1+1/SNR^2}\,(x_0^N - \mu_X), \tag{21}$$

$$\sigma^2_{Y|X^N=x_0^N} = \sigma^2_{\varepsilon^N} \tag{22}$$

$$= \left(1 - \frac{\rho^2}{1+1/SNR^2}\right)\sigma_Y^2, \tag{23}$$

where $\rho$ is the correlation between $X$ and $Y$, and $SNR = \sigma_X/\sigma_N$ is a signal-to-noise ratio, where $X$ is the signal and $N$ is the noise. Using correlation and signal-to-noise ratio to express the equations is inspired by Bowman et al. (2018). The equations (21) and (23) are very useful because they use parameters related to $X$, not $X^N$. The parameters $b_0$, $b_1$ and $\sigma^2_{\varepsilon^N}$ can thus be

computed using the sample of $X$ noiseless. This formalisation is possible thanks to the theory of measurement error models Fuller (2009). Using Eq. 20 and Eq. 22, the *PI* of $Y|X^N = x_0^N$ described Eq. 18 can be rewrite as:

$$PI_{1-\alpha}(Y|X^N = x_0^N) = \left[b_0 + b_1 x_0^N \pm z\,\sigma_{\varepsilon^N}\right]. \tag{24}$$

These results are interpreted mathematically and graphically using the same synthetic case study as before, detailed in appendix C. Fig. 7.a shows the *PI* of $Y$ constrained by a noisy observation, $PI_{1-\alpha}(Y|X^N = x_0^N)$ (green interval). It also shows how

it is constructed by plotting the linear regression $y = b_0 + b_1 x$ (green line) and its error $\sigma_{\varepsilon^N}$ (green shade). For comparison, it also shows the linear regression and its error obtained when the observational noise is neglected, as in the previous section, in red. Fig. 7.b compares $PI_{1-\alpha}(Y|X^N = x_0^N)$ (green) with the *PI* of $Y$ constrained by a noiseless observation (red) and unconstrained (black), respectively $PI_{1-\alpha}(Y|X = x_0)$ and $PI_{1-\alpha}(Y)$.

The expression of the *PI* of $Y$ constrained by a noisy observation ($PI(Y|X^N = x_0^N)$) has a form close to that where the

observational noise was neglected in the previous section ($PI(Y|X = x_0)$). As before, the expectation of $Y$ constrained is directly the real-world observation fed into the regression (see Eq. (20)), and the variance of $Y$ constrained is the variance of the regression error (see Eq. (22)). The constraint corrects the expectation (see Eq. (21)) and reduces the variance (see Eq. (23)). However, the difference between including or not including the observational noise (difference between green and red in Fig. 7) lies in a term called here the attenuation coefficient: $1/(1+1/SNR^2)$. The slope considering the observational noise, $b_1$, is

attenuated compared to the slope neglecting the observational noise, $a_1$: $b_1 = a_1/(1+1/SNR^2)$. The larger the observational noise, the larger the attenuation. This is illustrated in Fig. 7.a, where the linear relationship is stronger when the observational





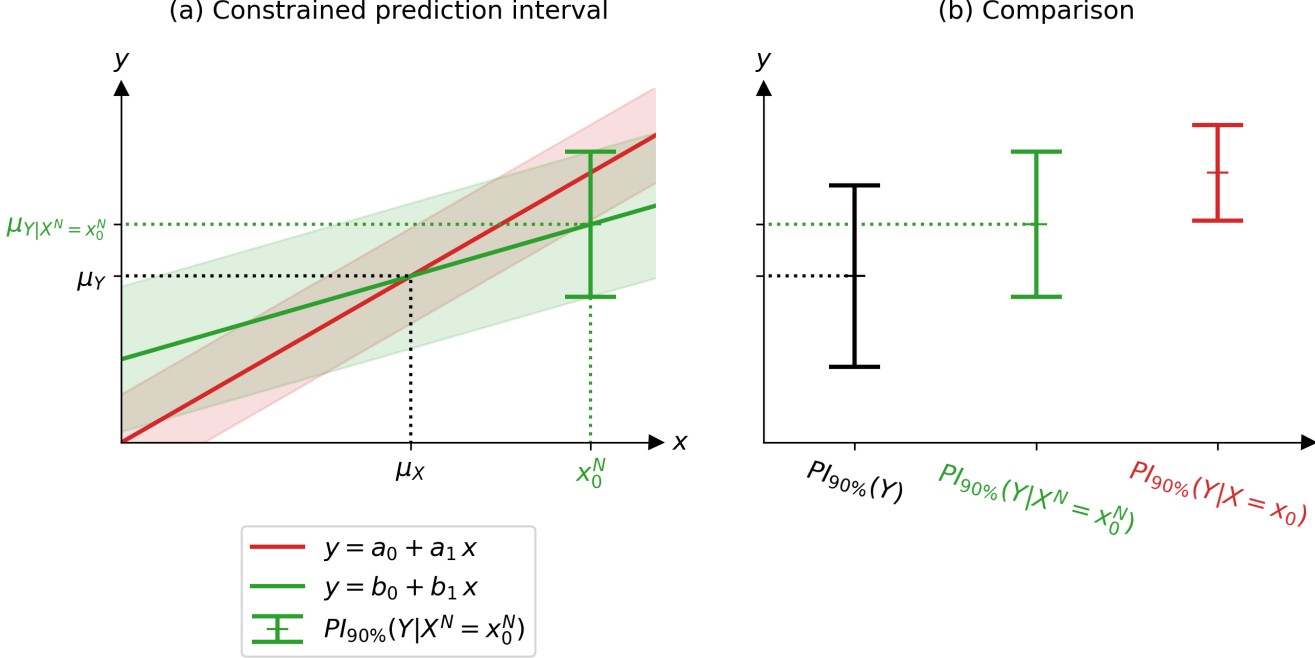

**Figure 7.** (a) Example showing in green the 90% probability interval (*PI*) of the projected variable $Y$ constrained by the noisy observation $x_0^N$, as described by Eq. (24). The green colour corresponds to the case where the observation is noisy, while the red colour corresponds to the case where the observation is considered noiseless, as in the previous section. (b) Comparison between the 90% *PI* of $Y$ unconstrained (black), constrained by a noisy observation (green) and constrained by a noiseless observation (red), corresponding respectively to Eq. (2) vs. Eq. (24) vs. Eq. (14). The values of means, variances, *etc* are given in appendix C.

noise is neglected (red) than when it is included (green). In this example, there is as much signal as noise ($SNR = 1$). The attenuation coefficient is therefore 50%. In reality, depending on the application, the observational noise can be very small, although it is difficult to estimate, especially for low frequency internal variability, which can lead to serious overconfidence

(Bonnet et al., 2021). This attenuation coefficient, $1/(1+1/SNR^2)$, weakens both the expectation correction and the variance reduction, as described in Eq. (21) and Eq. (23) respectively. This highlights the need to account for observational noise, otherwise the *PI* of $Y$ constrained will be overconfident with too strong an expectation correction.

The use of the *PI* of $Y|X^N = x_0^N$ described by Eq. (24) requires knowledge of the parameters $b_0$, $b_1$ and $\sigma_{\varepsilon^N}$, which are unknown in reality. As in the previous section, it is used an ensemble of $M$ climate model projections to estimate them. The

estimators of $a_0$, $a_1$ and $\sigma_{\varepsilon^N}$ are given in appendix table A2. Using them, it is shown in appendix G that the confidence interval (*CI*) of $Y$ constrained by a noisy observation is:

$$CI_{1-\alpha}(Y|X^N = x_0^N) = \left[ \hat{b}_0 + \hat{b}_1 x_0^N \pm t^{M-2} \hat{\sigma}_{\varepsilon^N} \sqrt{1 + \frac{1}{M} + \frac{(x_0^N - \hat{\mu}_X)^2}{M(\hat{\sigma}_X^2 + \sigma_N^2)}} \right]. \tag{25}$$





When $X$ is multivariate, its expression is:

$$CI_{1-\alpha}(Y|X^N = x_0^N) = \left[ \hat{b}_0 + \hat{b}_1^T x_0 \pm t^{M-1-p} \hat{\sigma}_{\varepsilon^N} \sqrt{1 + \frac{1}{M} + (x_0 - \hat{\mu}_X)^T \frac{(\hat{\Sigma}_X + \Sigma_N)^{-1}}{M} (x_0 - \hat{\mu}_X)} \right],$$
(26)

where $p$ is the number of features in $X$ and $\Sigma_X$ and $\Sigma_N$ are the variance-covariance matrices of $X$ and $N$, respectively. The confidence interval of $Y$ constrained by a noisy observation ($CI_{1-\alpha}(Y|X^N = x_0^N)$) described in Eq. (26) is the statistical model called "CLimate variable confidence Interval of Multivariate Linear Observational COnstraint" (*ClimLoco1.0*). To illustrate these mathematical results, Fig. 8 shows two realisations of *ClimLoco1.0*, one realisation from the sample $(X_1, Y_1), ..., (X_M, Y_M)$ of size $M = 5$ and one realisation from the sample of size $M = 30$, taken from the same synthetic example as before. In

Fig. 8.a., each sample realisation gives a different realisation of the linear relationship $y = \hat{b}_0 + \hat{b}_1 x$ (green line) and the confidence interval constrained by a noisy observation described in Eq. 25 (green interval). The green shading represents the $CI_{90\%}(Y|X^N = x_0^N)$ obtained for different positions of the observation $x_0^N$. For comparison, this Fig. 8.a shows in red the $CI_{90\%}(Y|X = x_0)$ for different positions of $x_0$ (red shade). This enables to compare the difference in intervals width and centre if the observational noise is considered or neglected. Fig. 8.b compares the *CI* of $Y$: unconstrained ($CI_{90\%}(Y)$, in black),

constrained by a noiseless observation ($CI_{90\%}(Y|X = x_0)$, in red) and constrained by a noisy observation (*ClimLoco1. 0*, $CI_{90\%}(Y|X^N = x_0^N)$, in green).

When comparing the *CI* of $Y$ constrained by a noisy vs. noiseless observation, green vs. red in Fig. 8.b, it is founded the same previous conclusions as when comparing *PI* of $Y$ constrained by noisy vs. noiseless observation: there is a decrease of reduction in uncertainty (interval width) and of correction of best guess (interval centre). In other words, observational noise

weakens the constraint. When comparing the two rows of Fig. 8, corresponding a small (first row) and large sample (second row), the large sample leads to narrower *CI*. The *CI* is more precise when estimated on more data. This is visible in all three expressions of the *CI* discussed in this article with the effect of the term $M$. Moreover, this synthetic example uses a strong observational noise ($SNR = 1$). Combined with a small sample (first row of Fig. 8), this tends to make the $CI(Y|X^N = x_0^N)$ large, which means the uncertainty is large. Therefore, the $CI(Y|X^N = x_0^N)$ is larger than the $CI(Y)$ in the second row:

the constraint has not reduced the uncertainty, which is surprising. However, this is an extreme case, combining both high observational noise and small sample size. In summary, low observational noise combined with a high correlation between $X$ and $Y$ leads to a strong constraint, which means a strong best guess correction (centre of the confidence interval) and a strong uncertainty reduction (width of the confidence interval). Uncertainty is also affected by sample size: the larger the sample size, the greater the uncertainty reduction. The best guess correction is also affected by the distance between the observation and the

multi-model mean ($x_0 - \hat{\mu}_X$), which is called in this article the "multi-model bias". The larger the bias, the larger the correction.

The main contributions of this section are to provide the statistical model *ClimLoco1.0*, the confidence interval of the projected variable constrained by a noisy observation, to express and illustrate it graphically as an attenuated linear regression, and to highlight the need to take this observational noise into account and to have a sample size as large as possible. Figure 9 is proposed as an illustrative summary of a comparison of the *CI* of $Y$ unconstrained, of $Y$ constrained by a noiseless observation,

and of $Y$ constrained by a noisy observation. This figure is built using synthetic data detailed in appendix C.







**Figure 8.** Synthetic example showing (a) (the first column) two realisations of the confidence interval (*CI*) of $Y$ constrained by a noisy (respectively noiseless) observation, shown in green (respectively red), given with 90% confidence. The shades correspond to the intervals obtained from different positions of the observation. The first (respectively second) row corresponds to a realisation of a sample of size $M = 5$ (respectively $M = 30$). (b), the second column, compares the *CI*s of $Y$ unconstrained (black) vs. constrained by a noiseless observation (red) vs. constrained by a noisy observation (green), corresponding respectively to Eq. (6) vs. Eq. (25) vs. Eq. (15). The details of the data simulation are given in appendix C.

## 5 Discussion

In this section, the results of this article are compared with those of some of the most widely used approaches in the observational constraint literature: (a) Ribes et al. (2021) and Bowman et al. (2018), and (b) Cox et al. (2018).

(a) Both Ribes et al. (2021) and Bowman et al. (2018) use statistical approaches to constrain $Y$ by real-world observations.
One can demonstrate (not shown here) that these two articles give equivalent expressions of the expectation and variance of $Y|X^N = x_0^N$. The main difference between them is that the first article considers the variables $X$ and $Y$ as univariate and



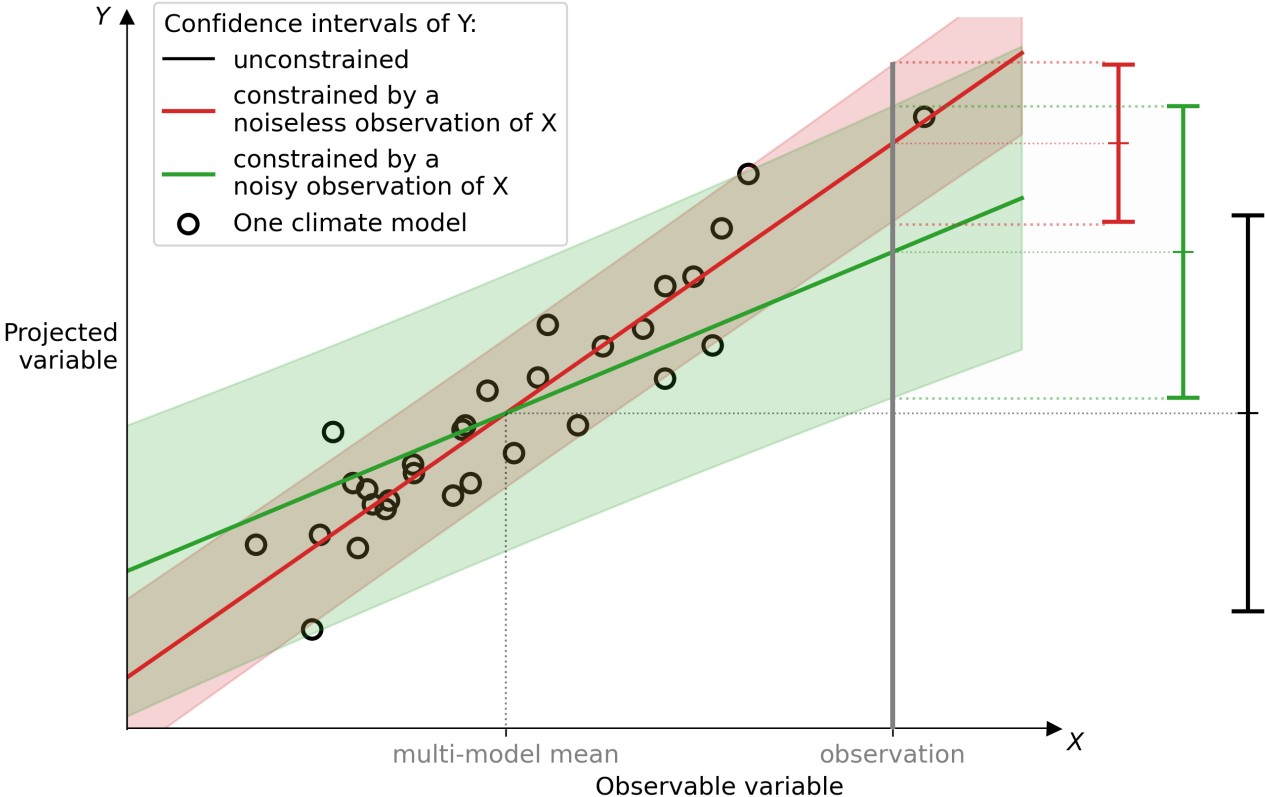

**Figure 9.** Graphical representation of the effect of considering observational noise in a linear observational constraint (*OC*). (i) In red, the observational noise is neglected. (ii) In green, the observational noise is considered, which is more rigorous. The green confidence interval (*CI*) corresponds to the statistical model *ClimLoco1.0* presented in this article. (i) When observational noise is neglected, a linear relationship is defined between a past observable variable $X$ and a future variable $Y$ using an ensemble of climate models (black circles). The slope and error of the relationship between $X$ and $Y$ are shown as the red line and shading. A real-world observation of $X$ is then fed into the linear relationship to obtain the *CI* of $Y$ constrained (red interval). Compared to the *CI* of $Y$ unconstrained (black interval), the *CI* of $Y$ constrained (red interval) has a reduced uncertainty (interval width) and a corrected best guess (interval centre). The intensity of the best guess correction (between unconstrained and constrained) depends on the the slope between $X$ and $Y$, and the difference between the multi-model mean and the observation (the "multi-model bias"). (ii) However, it does not take into account the uncertainty associated with the real-world observation. When taken into account, observational noise reduces the slope (green line) of the linear relationship and increases its error (green shade). Consequently, the *CI* of $Y$ constrained by a noisy observation (green interval) has less uncertainty reduction and less best guess correction than the *CI* of $Y$ constrained by a noiseless observation (red interval). All three *CI*s use a 90% confidence level.

the second as multivariate, respectively. It can be found that these articles have the same expressions for the expectation and variance of $Y|X^N = x_0^N$ as those obtained in Eq. (21) and Eq. (23). This means that the approaches of both Ribes et al. (2021) and Bowman et al. (2018) are equivalent to using a linear regression model (multivariate and univariate, respectively). Note that



this regression is corrected by the observational noise, as seen in the previous section. This is an important result for interpreting these methods using linear regression, as is done in our article. Furthermore, an important caveat to this equivalence is that there is a well-known risk of overfitting when using multivariate linear regression, i.e. learning incorrect relationships between features by over-fitting the data. This risk is greater when the number of variables is large and the number of climate models used to learn the regression is small. The multivariate method developed by Ribes et al. (2021) therefore presents a risk of overlearning.

Furthermore, the articles by Bowman et al. (2018) and Ribes et al. (2021) only gave the theoretical expressions for the expectation and variance of $Y|X^N = x_0^N$. These theoretical values are in reality unknown. They did not give details of the exact expression of the estimates, which, as previously seen using confidence intervals, leads to a higher uncertainty due to the limited sample size. This can be neglected when the sample size is very large, but is very important to take into account when it is small, as shown in the previous section (see Eq. 6). In climate science, sample sizes are usually small (especially if only considering high resolution models (Bauer et al., 2021)), so we argue here that this uncertainty must be included in the estimates.

(b) When referring to observational constraints, an often quoted figure comes from Eyring et al. (2019), Box 1. This method is used in several papers, e.g. Bracegirdle and Stephenson (2012), Brient (2020). Tthis figure is interpreted here using the well-known paper of Cox et al. (2018). Our approach leads to a different expression of the expectation and variance of $Y$ constrained by a noisy observation than the approach of Cox et al. (2018), for which we argue our disagreement here. Cox et al. (2018) studies the distribution of $Y$ using the law of total probability (see Eq. (15) in Cox et al. (2018)). Written differently, using the laws of total expectation and variance, the expectation and variance of this distribution can be expressed as:

$$\mathbb{E}[Y] = \mathbb{E}[\mathbb{E}[Y|X]], \tag{27}$$

$$\mathbb{V}(Y) = \mathbb{E}[\mathbb{V}[Y|X]] + \mathbb{V}[\mathbb{E}[Y|X]]. \tag{28}$$

Using a linear regression between $Y$ and $X$, noted $Y = a_0 + a_1 X + \varepsilon$, it gives:

$$\mathbb{E}[Y] = a_0 + a_1\mathbb{E}[X], \tag{29}$$

$$\mathbb{V}(Y) = \mathbb{V}(\varepsilon) + a_1^2\mathbb{V}(X). \tag{30}$$

Cox et al. (2018) assumes that $X$ follows a distribution centered around the observation ($\mathbb{E}[X] = x_0^N$) and of variance the observational noise variance ($\mathbb{V}(X) = \sigma_N^2$). Consequently,

$$\mathbb{E}[Y] = a_0 + a_1 x_0^N, \tag{31}$$

$$\mathbb{V}(Y) = \mathbb{V}(\varepsilon) + a_1^2 \sigma_N^2. \tag{32}$$

This corresponds to the figure in Eyring et al. (2019), Box 1: the best guess is directly the observation fed into the regression ($a_0 + a_1 x_0$), and the total uncertainty is the regression uncertainty ($\mathbb{V}(\varepsilon)$) plus the observational uncertainty fed into the regression ($a_1^2 \sigma_N^2$).





We suggest that there are two main problems with this approach. Firstly, Cox et al. (2018) uses two different distributions of the same variable $X$, one from the climate models to learn the linear relationship $Y = a_0 + a_1 X + \varepsilon$ and one from the noisy observation. But the climate models have a different $X$ distribution than the observation: $\mathbb{E}[X] = \mu_X \neq x_0^N$ and $\mathbb{V}(X) = \sigma_X^2 \neq \sigma_N^2$. The variable $X$ cannot have two different expectations and two different variances, the equations (31) and (32) written

above are incorrect from our point of view. It is necessary to separate the variable $X$, whose distribution is given by the climate models, from the variable $X^N$, which is observed from the real-world with an observational noise, as is done in this article. Secondly, the constrained variable should be noted $Y|X^N = x_0^N$, not just $Y$. This has a major effect on the resulting equations. Indeed, the equations Eq. (29) and Eq. (30) are correct, but do not constrain either the expectation or the variance of $Y$.

    This conclusion is consistent with Hall et al. (2019), who states that "care must be taken to characterise the uncertainty

in the observational values of the X variable. The translation of observed X-values into predicted Y-values is not trivial. It is certainly not as simple as finding the intersection of the most likely value of observed X and the regression line relating Y to X and 'reading' the predicted Y value. Instead, both observed X and predicted Y must be treated statistically." The clarification proposed here may help to move into this direction.

## 6    Conclusions

A confidence interval of future climate, i.e. a best guess of future climate with uncertainty given at a confidence level, can be obtained from an ensemble of climate model projections. However, the large dispersion between climate model projections makes this interval large, and consequently the future climate very uncertain. To refine it, methods called observational constraints (*OC*) combine climate model projections with some real-world observations (*cf IPCC* 2021). These methods are now increasingly used (O'Reilly et al., 2024), even by potential stakeholders at the national level (e.g. Ribes et al., 2022). They

therefore deserve to be rigorously described in their assumptions and mathematical description. However, there are many challenges in dealing with the literature of *OC*. There is a wide variety of *OC* methods, which are sometimes difficult to reproduce and may lack mathematical details, which are usually limited to the use of a single variable observable to contrain, and which do not strictly use confidence intervals, which are essential to correctly define uncertainty.

    To address these challenges, this article proposes a new (*1.0*) statistical method called *ClimLoco*, which stands for "CLimate

variable confidence Interval of Multivariate Linear Observational COnstraint". *ClimLoco1.0* describes the confidence interval of a projected variable constrained by a noisy observation using a multivariate linear framework. It is inspired by the theory of measurement error models from Fuller (2009). We found that constraining a projected variable have two effects: it corrects the best guess of the projected variable depending to the multi-model bias (difference between the multi-model mean and the real-world observation) and reduces the associated uncertainty.

Compared to the literature, *ClimLoco1.0* allows a more rigorous expression of uncertainty thanks to the use of confidence intervals. This takes into account the quality of the estimators of the best guess and the uncertainty of a projected variable, which depends in particular on the number of climate models used. We have therefore emphasised the need to have as large an ensemble of models as possible in order to obtain the most accurate estimates. In addition, *ClimLoco1.0* takes into account the





observational noise in a rigorous framework, which is important to correctly estimate the uncertainty. We find a new graphical
interpretation, *cf* Fig. 9, of the effect of observational noise, which weakens the constraint (less reduction of uncertainty and
less change in the best guess). This article is intended to be didactic, building the statistical model *ClimLoco1.0* step by step,
from the unconstrained case to the case constrained by noisy observations, and illustrating each step with univariate examples.

In addition, the results are compared with some of the most commonly used methods in the literature: "statistical" methods
(e.g. Bowman et al., 2018; Ribes et al., 2021), "linear regression" methods (e.g. Cox et al., 2018). There are strong similarities
between the statistical methods of Bowman et al. (2018) and Ribes et al. (2021) and the multivariate linear regression *OC*
developed in this article. We argue that there is an equivalence between their methods and a *multiple* linear regression. This
implies that the methods are subject to the risk of overfitting (i.e. learning incorrect relationships between features by over-
fitting the data). The use of methods to limit overfitting, such as ridge regression, seems to be a promising perspective in this
respect. However, since Bowman et al. (2018) and Ribes et al. (2021) did not use confidence intervals, they neglect the quality
of the estimators, which depends on the number of climate models considered. They therefore underestimate the uncertainty.
There is a major discrepancy between our method and that of Cox et al. (2018), which is now largely used for linear regression
*OC*s. We highlight problems in the underlying mathematics, and propose a new figure (Fig. 9), which may be more appropriate
than Fig. 1 from Eyring et al. (2019), to describe exactly how linear *OC* works in a geometric sense.

The statistical model *ClimLoco1.0* is an effort to better account for uncertainties and bring more clarity to *OC* methods.
However, there are still some challenges to overcome, which are interesting perspectives to build more advanced versions of
*ClimLoco1.0*. Firstly, *ClimLoco1.0* allows the internal variability of the observations to be considered as a source of noise.
However, the current version *1.0* is not able to take into account the internal variability present in climate models. This leads
to an underestimation of the uncertainty, the intensity of which depends on the variable considered. The inspiration of Olson
et al. (2018), which considers not only a value but a distribution from each climate model, seems to be a promising perspec-
tive. Secondly, *ClimLoco1.0* assumes that climate models are independent and equally plausible. These (false) assumptions
are refuted by methods that assign weights to climate models, e.g. Brunner et al. (2019). These methods are often used in the
literature on *OC* methods, but defining confidence intervals without the assumptions of independence and equal plausibility
using weighted samples is a challenge that *ClimLoco1.0* could improve. Thirdly, in this article, we have established an equiva-
lence between methods sometimes called "Bayesian" (Bowman et al., 2018; Ribes et al., 2021) and multiple linear regression.
Finding equivalences between other *OC* methods can be very useful to bring more clarity to the large literature of *OC* methods.
For example, Karpechko et al. (2013) succeeded in converting a linear regression into climate model weights, but neglected the
observational noise. Finally, *ClimLoco1.0* can be improved in many other ways: removing the assumption that variables have a
Gaussian distribution, constraining several variables at the same time, *etc*. This article presents the first version of *ClimLoco1.0*
which might be open to even further lines of improvements. We believe this might help to refine the future projection estimates
and lead to better adaptation plans.





*Code and data availability.* The python code used to generate the data and figures is available in a jupyter notebook script under https://doi.org/10.5281/zenodo.14679875 (Portmann, 2024) (last access: 17/01/2025).

## Appendix A: Summary

| $Y$ unconstrained $CI_{1-\alpha}(Y)$ | $\left[\hat{\mu}_Y \pm t^{M-1}\hat{\sigma}_Y\sqrt{1+\frac{1}{M}}\right]$ |
|---|---|
| $Y$ constrained by a noiseless observation $CI_{1-\alpha}(Y\|X=x_0)$ | $\left[\hat{a}_0 + \hat{a}_1 x_0 \pm t^{M-2}\hat{\sigma}_\varepsilon\sqrt{1+\frac{1}{M}+\frac{(x_0-\hat{\mu}_X)^2}{M\hat{\sigma}_X^2}}\right]$ if $X\in\mathbb{R}$ $\left[\hat{a}_0 + \hat{a}_1^T x_0 \pm t^{M-1-p}\hat{\sigma}_\varepsilon\sqrt{1+\frac{1}{M}+(x_0-\hat{\mu}_X)^T\frac{\hat{\Sigma}_X^{-1}}{M}(x_0-\hat{\mu}_X)}\right]$ if $X\in\mathbb{R}^p$ |
| $Y$ constrained by a noisy observation $CI_{1-\alpha}(Y\|X^N=x_0^N)$ | $\left[\hat{b}_0 + \hat{b}_1 x_0 \pm t^{M-2}\hat{\sigma}_{\varepsilon^N}\sqrt{1+\frac{1}{M}+\frac{(x_0-\hat{\mu}_X)^2}{M(\hat{\sigma}_X^2+\sigma_N^2)}}\right]$ if $X\in\mathbb{R}$ $\left[\hat{b}_0 + \hat{b}_1^T x_0 \pm t^{M-1-p}\hat{\sigma}_{\varepsilon^N}\sqrt{1+\frac{1}{M}+(x_0-\hat{\mu}_X)^T\frac{(\hat{\Sigma}_X+\Sigma_N)^{-1}}{M}(x_0-\hat{\mu}_X)}\right]$ if $X\in\mathbb{R}^p$ |

**Table A1.** Confidence intervals (*CI*) of $Y$ unconstrained, constrained by a noiseless observation, and constrained by a noisy observation. It is given, within each case, the results both when $X$ is univariate ($X\in\mathbb{R}$) and when $X$ is multivariate ($X\in\mathbb{R}^p$). Since the first case does not depend on $X$, it is the same whether $X$ is univariate or multivariate. The different estimators are listed in the table A2 and described in the main part of the article.





| $X$ univariate | $X$ multivariate |
|---|---|
| $\hat{\mu}_Y = \frac{1}{M} \sum_{i=1}^{M} Y_i$ | |
| $\hat{\mu}_X = \frac{1}{M} \sum_{i=1}^{M} x_i$ | |
| $\hat{\sigma}_Y^2 = \frac{1}{M-1} \sum_{i=1}^{M} (Y_i - \hat{\mu}_Y)^2$ | |
| $\hat{\sigma}_X^2 = \frac{1}{M-1} \sum_{i=1}^{M} (x_i - \hat{\mu}_X)^2$ | $\hat{\Sigma}_X = \frac{1}{M-1} \sum_{i=1}^{M} (x_i - \hat{\mu}_X)^T (x_i - \hat{\mu}_X)$ |
| $\hat{a}_0 = \hat{\mu}_Y - \hat{a}_1^T \hat{\mu}_X$ | |
| $\hat{a}_1 = \hat{Cov}(Y,X)/\hat{\sigma}_X^2$ | $\hat{a}_1^T = \hat{\Sigma}_{YX} \hat{\Sigma}_X^{-1}$ |
| $\hat{Cov}(Y,X) = \hat{\Sigma}_{YX} = \frac{1}{M-1} \sum_{i=1}^{M} (Y_i - \hat{\mu}_Y)(x_i - \hat{\mu}_X)^T$ | |
| $\hat{\sigma}_\varepsilon^2 = \frac{1}{M-1} \sum_{i=1}^{M} (Y_i - \hat{a}_0 - \hat{a}_1^T x_i)^2$ | |
| $\hat{b}_0 = \hat{\mu}_Y - \hat{b}_1^T \hat{\mu}_X$ | |
| $\hat{b}_1 = \hat{Cov}(Y,X)/(\hat{\sigma}_X^2 + \sigma_N^2)$ | $\hat{b}_1 = \hat{\Sigma}_{YX}(\hat{\Sigma}_X + \Sigma_N)^{-1}$ |
| $\hat{\sigma}_{\varepsilon N}^2 = \frac{1}{M-1} \sum_{i=1}^{M} (Y_i - \hat{b}_0 - \hat{b}_1 x_i)^2 + \hat{b}_1^2 \sigma_N^2$ | $\hat{\sigma}_{\varepsilon N}^2 = \frac{1}{M-1} \sum_{i=1}^{M} (Y_i - \hat{b}_0 - \hat{b}_1^T x_i)^2 + \hat{b}_1^T \Sigma_N \hat{b}_1$ |

**Table A2.** Estimators used in this article, given when $X$ is univariate ($X \in \mathbb{R}$) and when $X$ is multivariate ($X \in \mathbb{R}^p$).

## Appendix B: Confidence interval of $Y$

The goal of this appendix is to find the confidence interval of $Y$. In this purpose, it is assumed that $Y$ follows a Gaussian distribution: $Y \sim \mathcal{N}(\mu_Y, \sigma_Y^2)$. To estimate $\mu_Y$ and $\sigma_Y^2$, it is used a sample of $M$ random variables, denoted $(Y_1, ..., Y_M)$, given by an ensemble of $M$ climate models. These random variables are assumed to be independent and to follow the same law as $Y$. The expectation and variance classical estimators are:

$$\hat{\mu}_Y = \frac{1}{M} \sum_{i=1}^{M} Y_i \quad \text{and} \quad \hat{\sigma}_Y^2 = \frac{1}{M-1} \sum_{i=1}^{M} (Y_i - \hat{\mu}_Y)^2.$$





On the one hand, $E[\hat{\mu}_Y] = \frac{1}{M}\sum_{i=1}^{M} E[Y_i] = \frac{1}{M}\sum_{i=1}^{M}\mu_Y = \mu_Y$

and $Var(\hat{\mu}_Y) = \frac{1}{M^2}\sum_{i=1}^{M} Var(Y_i) = \frac{1}{M^2}\sum_{i=1}^{M}\sigma_Y^2 = \frac{\sigma_Y^2}{M}$

$$\implies \hat{\mu}_Y \sim \mathcal{N}(\mu_Y, \frac{\sigma_Y^2}{M})$$

$$\implies Y - \hat{\mu}_Y \sim \mathcal{N}(0, \sigma_Y^2(1+\frac{1}{M}))$$

$$\implies \frac{Y - \hat{\mu}_Y}{\sigma_Y\sqrt{1+\frac{1}{M}}} \sim \mathcal{N}(0,1)$$

On the other hand, $\hat{\sigma}_Y^2 = \frac{1}{M-1}\sum_{i=1}^{M}(Y_i - \hat{\mu}_Y)^2$

$$\implies (M-1)\frac{\hat{\sigma}_Y^2}{\sigma_Y^2} = \sum_{i=1}^{M}\frac{(Y_i - \hat{\mu}_Y)^2}{\sigma_Y^2}$$

As $\frac{(Y_i - \hat{\mu}_Y)}{\sigma_Y} \sim \mathcal{N}(0,1)$

then $(M-1)\frac{\hat{\sigma}_Y^2}{\sigma_Y^2} \sim \chi^2(M-1)$


$$\text{Or } \begin{cases} U \sim \mathcal{N}(0,1) \\ V \sim \chi^2(n) \\ U \perp\!\!\!\perp V \end{cases} \implies \frac{U}{\sqrt{V/n}} \sim St(n),$$

With $\perp\!\!\!\perp$ the sign of independence, and $St(n)$ the student distribution with $n$ degrees of freedom. Consequently, by noting $U = \frac{Y - \hat{\mu}_Y}{\sigma_\varepsilon\sqrt{1+\frac{1}{M}}}$ and $V = (M-1)\frac{\hat{\sigma}_Y^2}{\sigma_Y^2}$, this implies that: $\frac{U}{\sqrt{V/(M-1)}} = \frac{Y - \hat{\mu}_Y}{\hat{\sigma}_Y\sqrt{1+\frac{1}{M}}} \sim St(M-1)$.

The confidence interval is thus: $\left[\hat{\mu}_Y \pm t^{M-1}\hat{\sigma}_Y\sqrt{1+\frac{1}{M}}\right]$, where $t^{M-1}$ is the quantile of a Student with $M-1$ degrees of
freedom.

**Appendix C: Simulation of the synthetic example**

To illustrate the mathematical results, it is used the same synthetic example throughout the article. It is simulated two different realisations coming from two samples $(X_1, Y_1), ..., (X_M, Y_M)$, one with $M = 30$ and the other with $M = 5$. The random variables $X$ and $Y$ have a centred reduced normal distribution ($\mu_X = \mu_Y = 0$, $\sigma_X = \sigma_Y = 1$). The correlation between $X$ and





$Y$ is chosen as $\rho = 0.85$. The linear relation between $Y$ and $X$ is therefore defined by $Y = a_0 + a_1 X + \varepsilon$ with $a_0 = 0$ and $a_1 = 0.85$. It is simulated a realisation of the sample $(X_1, ..., X_M)$ and a realisation of the sample $(\varepsilon_1, ..., \varepsilon_M)$ with $M = 30$ values. Then a sample of $(Y_1, ..., Y_M)$ is obtained using the relation $Y = a_0 + a_1 X + \varepsilon$. This gives the realisation of the first sample of size $M = 30$. The realisation of the second sample of size $M = 5$ is obtained by taking the first 5 values. For the observation, it is used the value $x_0 = 2.2$ and the observational noise standard deviation is chosen as $\sigma_N = \sigma_X = 1$ (a signal-

to-noise ratio of 1). For the sake of the illustration, the figure 9 uses the same data with two different parameters: $x_0 = 3$ and $\rho = 0.9$.

**Appendix D: Probability interval of $Y | X = x_0$**

The goal of this section is to find the probability interval (*PI*) of $Y$ constrained by the observation $x_0$ of $X$. It is denoted $PI_{1-\alpha}(Y | X = x_0)$, and contains the values of $Y | X = x_0$ with a given probability $1 - \alpha$. To obtain this interval, it is used the

following Gaussian assumption: $Y | X = x_0 \sim \mathcal{N}(\mu_{Y|X=x_0}, \sigma^2_{Y|X=x_0})$. Under this assumption, the *PI* of $Y$ constrained can be written as:

$$PI_{1-\alpha}(Y | X = x_0) = [\mu_{Y|X=x_0} \pm z\,\sigma_{Y|X=x_0}] \tag{D1}$$

where $z$ is the quantile of order $1 - \alpha/2$ of a centred reduced normal distribution. To obtain the expressions of the parameters $\mu_{Y|X=x_0}$ and $\sigma_{Y|X=x_0}$, it is used a multiple linear regression framework:


$$Y = \mathbb{E}[Y|X] + \varepsilon \tag{D2}$$
with $\mathbb{E}[Y|X] = a_0 + a_1^T X$

with $Y \in \mathbb{R}$, $X \in \mathbb{R}^p$, $a_0 \in \mathbb{R}$ and $a_1 \in \mathbb{R}^p$ the coefficients of the regression of $Y$ on $X$, and $\varepsilon \in \mathbb{R}$ the regression error. Using the latter equation, it is established (solution of the least square) that $\varepsilon \perp\!\!\!\perp X, \mathbb{E}[\varepsilon] = 0, a_1^T = \Sigma_{YX}\Sigma_X^{-1}$ and $a_0 = \mu_Y - a_1^T \mu_X$. The terms $\mu_{Y|X=x_0} = \mathbb{E}[Y|X = x_0]$ and $\sigma^2_{Y|X=x_0} = \mathbb{V}(Y|X = x_0)$ are then expressed using this multiple linear regression.





$$\mathbb{E}[Y|X = x_0] = a_0 + a_1^T x_0 \tag{D3}$$

$$= \mu_Y + a_1^T(x_0 - \mu_X) \text{ because } a_0 = \mu_Y - a_1^T \mu_X$$

$$\implies \mathbb{E}[Y|X = x_0] = \mu_Y + \Sigma_{YX}\Sigma_X^{-1}(x_0 - \mu_X) \text{ because } a_1^T = \Sigma_{YX}\Sigma_X^{-1}$$

$$\mathbb{V}(Y|X = x_0) = \mathbb{V}((a_0 + a_1^T X + \varepsilon)|X = x_0)$$

$$= \mathbb{V}(\varepsilon|X = x_0)$$

$$\implies \mathbb{V}(Y|X = x_0) = \mathbb{V}(\varepsilon) \text{ because } \varepsilon \perp\!\!\!\perp X$$

$$= \mathbb{V}(Y - a_0 - a_1^T X)$$

$$= \mathbb{V}(Y) + \mathbb{V}(-a_1^T X) + 2Cov(Y, -a_1^T X)$$

$$= \mathbb{V}(Y) + a_1^T \mathbb{V}(X)a - 2Cov(Y, X)a$$

$$= \sigma_Y^2 + a_1^T \Sigma_X a - 2\Sigma_{YX}a$$

$$= \sigma_Y^2 + \Sigma_{YX}\Sigma_X^{-1}\Sigma_X\Sigma_X^{-1}\Sigma_{XY} - 2\Sigma_{YX}\Sigma_X^{-1}\Sigma_{XY} \text{ because } a_1^T = \Sigma_{YX}\Sigma_X^{-1}$$

$$\implies \mathbb{V}(Y|X = x_0) = \sigma_Y^2 - \Sigma_{YX}\Sigma_X^{-1}\Sigma_{XY}$$

When $X$ is univariate ($p = 1$), the results can be written:

$$\mathbb{E}[Y|X = x_0] = \mu_Y + \rho\frac{\sigma_Y}{\sigma_X}(x_0 - \mu_X)$$

$$\mathbb{V}(Y|X = x_0) = (1 - \rho^2)\sigma_Y^2$$

With $\rho = \frac{Cov(Y,X)}{\sigma_X\,\sigma_Y}$ the correlation between $X$ and $Y$. The *PI* can consequently be noted:

$$PI_{1-\alpha}(Y|X = x_0) = \left[a_0 + a_1^T x_0 \pm z\,\sigma_\varepsilon\right]$$

or else:

$$PI_{1-\alpha}(Y|X = x_0) = \left[\mu_Y + \Sigma_{YX}\Sigma_X^{-1}(x_0 - \mu_X) \pm z\sqrt{\sigma_Y^2 - \Sigma_{YX}\Sigma_X^{-1}\Sigma_{XY}}\right]$$

## Appendix E: Confidence interval of $Y|X = x_0$

The goal of this appendix is to find the confidence interval of $Y$ given an observation $x_0$ of $X$, named $CI(Y|X = x_0)$, using an ensemble of $M$ climate models. This ensemble yields a sample of $M$ pairs of random variables, denoted $(X_1, Y_1), ..., (X_M, Y_M)$. They are assumed to be independent and to follow the same law as $(X, Y)$, which is assumed to be Gaussian. The relationship between $X$ and $Y$ is assumed to be linear:

$$Y = \mathbb{E}[Y|X] + \varepsilon \tag{E1}$$

with $\mathbb{E}[Y|X] = a_0 + a_1^T X$



With $Y \in \mathbb{R}$, $X \in \mathbb{R}^p$. $a_0 \in \mathbb{R}$ and $a_1 \in \mathbb{R}^p$ the coefficients of the regression of $Y$ on $X$, and $\varepsilon \in \mathbb{R}$ the regression error. It is used the estimators of $a_0$, $a_1$,*etc* detailed in table A2. The properties of the estimators $\hat{a}_0$ and $\hat{a}_1$ are well established: $\mathbb{E}[\hat{a}_0] = a_0$, $\mathbb{E}[\hat{a}_1] = a_1$, $\mathbb{V}(\hat{a}_0) = \frac{\sigma_\varepsilon^2}{M}(1 + \hat{\mu}_X^T \hat{\Sigma}_X^{-1} \hat{\mu}_X)$, $\mathbb{V}(\hat{a}_1) = \frac{\sigma_\varepsilon^2}{M}\hat{\Sigma}_X^{-1}$, and $Cov(\hat{a}_0, \hat{a}_1) = -\frac{\sigma_\varepsilon^2}{M}\hat{\mu}_X^T \hat{\Sigma}_X^{-1}$.

On the one hand, $E[\hat{\mu}_{Y|X=x_0}] = E[\hat{a}_0 + \hat{a}_1^T x_0]$

$$= a_0 + a_1^T x_0$$

and $Var(\hat{\mu}_{Y|X=x_0}) = Var(\hat{a}_0 + \hat{a}_1^T x_0)$

$$= Var(\hat{a}_0) + Var(\hat{a}_1^T x_0) + 2Cov(\hat{a}_0, \hat{a}_1^T x_0)$$

$$= \frac{\sigma_\varepsilon^2}{M}(1 + \hat{\mu}_X^T \hat{\Sigma}_X^{-1} \hat{\mu}_X) + x_0^T \frac{\sigma_\varepsilon^2}{M}\hat{\Sigma}_X^{-1} x_0 - 2\frac{\sigma_\varepsilon^2}{M}\hat{\mu}_X^T \hat{\Sigma}_X^{-1} x_0$$

$$= \frac{\sigma_\varepsilon^2}{M}[(1 + \hat{\mu}_X^T \hat{\Sigma}_X^{-1} \hat{\mu}_X) + x_0^T \hat{\Sigma}_X^{-1} x_0 - 2\hat{\mu}_X^T \hat{\Sigma}_X^{-1} x_0]$$

$$= \frac{\sigma_\varepsilon^2}{M}(1 + (x_0 - \hat{\mu}_X)^T \hat{\Sigma}_X^{-1}(x_0 - \hat{\mu}_X))$$

$$\implies \hat{\mu}_{Y|X=x_0} \sim \mathcal{N}(a_0 + a_1^T x_0, \frac{\sigma_\varepsilon^2}{M}(1 + (x_0 - \hat{\mu}_X)^T \hat{\Sigma}_X^{-1}(x_0 - \hat{\mu}_X)))$$

Or $Y|_{X=x_0} \sim \mathcal{N}(a_0 + a_1^T x_0, \sigma_\varepsilon^2)$

$$\implies Y|_{X=x_0} - \hat{\mu}_{Y|X=x_0} \sim \mathcal{N}(0, \sigma_\varepsilon^2(1 + \frac{1}{M} + (x_0 - \hat{\mu}_X)^T \frac{\hat{\Sigma}_X^{-1}}{M}(x_0 - \hat{\mu}_X)))$$

$$\implies U \sim \mathcal{N}(0,1) \text{ with } U = \frac{Y|_{X=x_0} - \hat{\mu}_{Y|X=x_0}}{\sigma_\varepsilon \sqrt{1 + \frac{1}{M} + (x_0 - \hat{\mu}_X)^T \frac{\hat{\Sigma}_X^{-1}}{M}(x_0 - \hat{\mu}_X)}}$$

On the other hand, noting $V = (M - 1 - p)\frac{\hat{\sigma}_\varepsilon^2}{\sigma_\varepsilon^2}$

$$= \sum_{1=1}^{M} \frac{(Y_i - \hat{a}_0 - \hat{a}_1^T X_i)^2}{\sigma_\varepsilon^2}$$

As $\frac{Y_i - \hat{a}_0 - \hat{a}_1^T X_i}{\sigma_\varepsilon} \sim \mathcal{N}(0,1)$

Then $V \sim \chi^2(M - 1 - p)$

Or $\begin{cases} U \sim \mathcal{N}(0,1) \\ V \sim \chi^2(n) \\ U \perp\!\!\!\perp V \end{cases} \implies \frac{U}{\sqrt{V/n}} \sim St(n),$



$$\implies \frac{U}{\sqrt{V/(M-1)}} = \frac{Y|_{X=x_0} - \hat{\mu}_{Y|X=x_0}}{\hat{\sigma}_\varepsilon \sqrt{1 + \frac{1}{M} + (x_0 - \hat{\mu}_X)^T \frac{\hat{\Sigma}_X^{-1}}{M}(x_0 - \hat{\mu}_X)}} \sim St(M-1-p).$$ The confidence interval of $Y$ constrained is consequently:

$$CI_{1-\alpha}(Y|X=x_0) = \left[ \hat{\mu}_{Y|X=x_0} \pm t^{M-1-p} \hat{\sigma}_\varepsilon \sqrt{1 + \frac{1}{M} + (x_0 - \hat{\mu}_X)^T \frac{\hat{\Sigma}_X^{-1}}{M}(x_0 - \hat{\mu}_X)} \right]$$

In univariate ($p=1$), this gives: $CI_{1-\alpha}(Y|X=x_0) = \left[ \hat{\mu}_{Y|X=x_0} \pm t^{M-2} \hat{\sigma}_\varepsilon \sqrt{1 + \frac{1}{M} + \frac{(x_0 - \hat{\mu}_X)^2}{M \hat{\sigma}_X^2}} \right]$

## Appendix F: Probability interval of $Y|X^N = x_0^N$

The goal of this section is to find the probability interval (*PI*) of $Y$ constrained by the noisy observation $x_0^N$ of $X^N$. It is denoted $PI_{1-\alpha}(Y|X^N = x_0^N)$, and contains the values of $Y|X^N = x_0^N$ with a given probability $1-\alpha$. To obtain this interval, it is used the following Gaussian assumption: $Y|X^N = x_0^N \sim \mathcal{N}(\mu_{Y|X^N=x_0^N}, \sigma^2_{Y|X^N=x_0^N})$. Under this assumption, the *PI* of $Y$ constrained can be written as:

$$PI_{1-\alpha}(Y|X=x_0) = \left[ \mu_{Y|X^N=x_0^N} \pm z\, \sigma_{Y|X^N=x_0^N} \right],$$

where $z$ is the quantile of order $1-\alpha/2$ of a centred reduced normal distribution. To obtain the expressions of the parameters $\mu_{Y|X^N=x_0^N}$ and $\sigma_{Y|X^N=x_0^N}$, it is used a multiple linear regression framework:

$$Y = \mathbb{E}[Y|X^N] + \varepsilon^N,$$

with $\mathbb{E}[Y|X^N] = b_0 + b_1^T X^N,$

(F1)

with $Y \in \mathbb{R}$, $X^N \in \mathbb{R}^p$, $b_0 \in \mathbb{R}$ and $b_1 \in \mathbb{R}^p$ the coefficients of the regression of $Y$ on $X^N$, and $\varepsilon^N \in \mathbb{R}$ the regression error. Using the same methodology than in the section D, it can demonstrate that:

$$\mathbb{E}[Y|X^N = x_0^N] = b_0 + b_1^T x_0^N$$
$$= \mu_Y + \Sigma_{YX^N} \Sigma_{X^N}^{-1} (x_0 - \mu_X)$$
$$\mathbb{V}(Y|X^N = x_0^N) = \mathbb{V}(\varepsilon^N)$$
$$= \sigma_Y^2 - \Sigma_{YX^N} \Sigma_{X^N}^{-1} \Sigma_{X^N Y}$$

To link $X^N$ and $X$, the noisy and noiseless versions of $X$, it is used the noise model defined in Bowman et al. (2018):

$X^N = X + N$, with $N \sim \mathcal{N}(0, \Sigma_N)$

As the observational noise $N$ is unrelated to the climate models, $N$ is independent of $X$ and $Y$. Consequently, $\Sigma_{X^N} = \Sigma_X + \Sigma_N$ and $\Sigma_{YX^N} = Cov(Y, X + N) = Cov(Y, X) = \Sigma_{YX}$. Thus, the previous equations can be written:

$$\mathbb{E}[Y|X^N = x_0] = \mu_Y + \Sigma_{YX}(\Sigma_X + \Sigma_N)^{-1}(x_0^N - \mu_X)$$
$$\mathbb{V}(Y|X^N = x_0) = \sigma_Y^2 - \Sigma_{YX}(\Sigma_X + \Sigma_N)^{-1}\Sigma_{XY}$$





In univariate ($p = 1$), this gives:

$$\mathbb{E}[Y|X^N = x_0] = \mu_Y + \frac{Cov(Y,X)}{\sigma_X^2 + \sigma_N^2}(x_0^N - \mu_X)$$

$$\mathbb{V}(Y|X^N = x_0) = \sigma_Y^2 - \frac{Cov^2(Y,X)}{\sigma_X^2 + \sigma_N^2}$$

Using the correlation $\rho = \frac{Cov(Y,X)}{\sigma_X \sigma_Y}$ and the signal to noise ratio $SNR = \frac{\sigma_X}{\sigma_N}$, this gives:

$$\mu_{Y|X^N = x_0^N} = \mu_Y + \rho \frac{\sigma_Y}{\sigma_X} \frac{1}{1 + 1/SNR^2}(x_0^N - \mu_X)$$

$$\sigma_{Y|X^N = x_0^N}^2 = (1 - \frac{\rho^2}{1 + 1/SNR^2})\sigma_Y^2$$

The prediction interval of $Y$ constrained by a noisy observation can thus be written as:

$$PI_{1-\alpha}(Y|X^N = x_0^N) = \left[ b_0 + b_1^T x_0 \pm z\,\sigma_{\varepsilon^N} \right]$$

or else:

$$PI_{1-\alpha}(Y|X^N = x_0^N) = \left[ \mu_Y + \Sigma_{YX}(\Sigma_X + \Sigma_N)^{-1}(x_0 - \mu_X) \pm z\sqrt{\sigma_Y^2 - \Sigma_{YX}(\Sigma_X + \Sigma_N)^{-1}\Sigma_{XY}} \right]$$

## Appendix G: Confidence interval of $Y|X^N = x_0^N$

The goal of this appendix is to find the confidence interval of $Y$ given an observation $x_0^N$ of $X^N$, $CI(Y|X^N = x_0^N)$, using an ensemble of $M$ climate models. This ensemble yields a sample of $M$ pairs of random variables, denoted $(X_1, Y_1), ..., (X_M, Y_M)$. They are assumed to be independent and to follow the same law as $(X, Y)$, which is assumed to be Gaussian. The relationship 575 between $X^N$ and $Y$ is assumed to be linear:

$$Y = \mathbb{E}[Y|X^N] + \varepsilon^N$$

with $\mathbb{E}[Y|X^N] = b_0 + b_1^T X^N$

(G1)

With $Y \in \mathrm{IR}$, $X^N \in \mathrm{IR}^p$. $b_0 \in \mathrm{IR}$ and $b_1 \in \mathrm{IR}^p$ the coefficients of the regression of $Y$ on $X^N$, and $\varepsilon^N \in \mathrm{IR}$ the regression error. Based on the same methodology than previously E, the confidence interval of $Y$ constrained by a noisy observation is:

$$CI_{1-\alpha}(Y|X^N = x_0^N) = \left[ \hat{b}_0 + \hat{b}_1^T x_0 \pm t^{M-1-p} \hat{\sigma}_{\varepsilon^N} \sqrt{1 + \frac{1}{M} + (x_0 - \hat{\mu}_{X^N})^T \frac{\Sigma_{X^N}^{-1}}{M}(x_0 - \hat{\mu}_{X^N})} \right]$$

Using the noise model that link $X^N$ to $X$ (Bowman et al., 2018): $X^N = X + N$, with $N \sim \mathcal{N}(0, \Sigma_N)$ and $N \perp\!\!\!\perp X$, then $\hat{\mu}_{X^N} = \hat{\mu}_X$ and $\Sigma_{X^N}^{-1} = \Sigma_X^{-1} + \Sigma_N^{-1}$. The confidence interval of $Y|X^N = x_0^N$ can therefore be written:

$$CI_{1-\alpha}(Y|X^N = x_0^N) = \left[ \hat{b}_0 + \hat{b}_1^T x_0 \pm t^{M-1-p} \hat{\sigma}_{\varepsilon^N} \sqrt{1 + \frac{1}{M} + (x_0 - \hat{\mu}_X)^T \frac{(\Sigma_X + \Sigma_N)^{-1}}{M}(x_0 - \hat{\mu}_X)} \right]$$



The estimators of $b_0$, $b_1$, *etc* are detailed in table A2. In univariate ($p = 1$), the confidence interval of $Y|X^N = x_0^N$ can be written:

$$CI_{1-\alpha}(Y|X^N = x_0^N) = \left[ \hat{b}_0 + \hat{b}_1^T x_0 \pm t^{M-2} \hat{\sigma}_{\varepsilon^N} \sqrt{1 + \frac{1}{M} + \frac{(x_0 - \hat{\mu}_X)^2}{M(\hat{\sigma}_X^2 + \sigma_N^2)}} \right]$$

*Author contributions.* V. P. prepared the manuscript with contributions from all co-authors. The methodology were conducted by V. P. and M. C. with contribution from D. S., who carried the supervision. The code and figures were produced by V. P.

*Competing interests.* The authors declare that they have no conflict of interest.

*Acknowledgements.* We would like to thank Aurelien Ribes and Saïd Qasmi for helpful discussions during the development of the methodology. This work was supported by the Institut national de recherche en informatique et en automatique (INRIA), the TipESM (Grant No.
101137673) and the Blue-Action (Grant No. 727852) projects funded by the European Union's Horizon 2020 research and innovation programme.



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
