# Peer review of "ClimLoco 1.0: CLimate variable confidence Interval of Multivariate Linear Observational COnstraint"

_EGUsphere, 2025_

## Author Response (AR1)

We thank both reviewers for their comments about the article. We appreciate their constructive and detailed feedback. We provide below our responses to the reviewers' comments. Those comments are noted in *italics* while our responses are in regular font.

Following these comments, the main additions to the article can be summed up as:

- a figure describing the flowchart of the article in the introduction,
- a section that summarises and discuss the assumptions used in ClimLoco1.0,
- a discussion comparing ClimLoco to other non-linear multivariate methods from the literature,
- an appendix section that provides the key statistical concepts used in the article,
- an appendix section that illustrates the use of ClimLoco by a case study. It is also used to perform a sensitivity test and a comparison with other existing methods.

**Reviewer 1:**

I really like this paper. The authors focus on something that is known but perhaps not as widely appreciated as it could be – observations have errors, and imposing tight constraints on models assuming the observations are "truth" is not appropriate. I particularly like how the authors have essentially automated this process in a software package. This is a valuable tool for computing observational constraints. I particularly appreciate the authors' careful treatment of statistics, for example Figure 2 and the surrounding discussion.

My perspective is that of someone who is quite familiar with CMIP6, but I will admit that I am not well qualified to provide an in-depth review of the formulas and equations provided. They seem to be reasonable based on my knowledge of how t-distributions work, but my hope is that these are covered more in depth by another reviewer who spends a lot more time thinking about such things.

We would like to thank the reviewer for their useful comments on the manuscript, and for the time they spent helping us to improve it.

My biggest issue with this paper is stated on line 74 and then used throughout the rest of the paper, namely that the assumed underlying distribution of the projected variable is random. This is not always true (as the authors know), and for some important cases it is demonstrably false. One example is the hot bias of some of the CMIP6 models that received quite a bit of attention recently. Another is that this tool is essentially not usable for precipitation, which is known to (more or less) have an extreme value distribution. I realize that modifying this tool so that one can specify prior distributions rather than assume them would be far too much work for the scope of this paper. But I would like to see two things in the paper that address this comment: 1) An acknowledgment of this point and appropriate caveats in the paper, particularly in Section 2. Line 457 is the only place I saw this explicitly mentioned, and that makes it seem kind of buried. 2) A brief discussion as to either how one could specify prior distributions (and the information needed) or an express acknowledgment of the limitations of this tool. There is a hint of this on lines 450-452, but I don't think that's sufficient.

In response to this comment and a similar one from the second reviewer, we added a section called 'Discussion of the assumptions' after section 4. This new section summarises all the assumptions used in ClimLoco1.0 and discusses their implications, proposing future potential improvement. Regarding the Gaussian assumption, we added the following paragraph:

"The assumption that the distributions are Gaussian is clearly recognized as potentially questionable, e.g. in regard to precipitation. If the distribution is not centred, the confidence interval should not be centred either. If the distribution has significant tailed areas, the limits

of the confidence interval must be further apart. The greater the difference between the distribution and a Gaussian distribution, the less accurate ClimLoco1.0 will be in estimating the limits of the confidence interval. However, we did not estimate here whether this impact is significant or negligible. To address this issue in future development of ClimLoco1.0 using non-Gaussian distributions, we recommend employing a bootstrap method to empirically derive a confidence interval. Bootstrapping involves repeatedly resampling the dataset with replacement to create different sub-datasets. Each sub-dataset yields a different observational constraint result. These distributions are then used to compute a confidence interval. However, in this case, no analytical expression of the confidence interval can be derived since this remains an empirical approach."

I would also appreciate a bit more description of the software aspects of this tool. For example, what language it's written in, package dependencies, a clear description of inputs/outputs, etc. In one example, as I was reading Section 2 (around line 85), I was wondering if alpha is user-specifiable. Those sorts of things would be good to know. I did look at the python notebook available, but the readme is rather terse, and the code could be better commented, so it was difficult to discern some of these answers on my own.

We modified the GitHub repository to clearly indicate all the necessary information in the 'readme' file (installation and use of the software, package dependencies, inputs/outputs, etc.). We modified and commented the code to make it easier to understand and be used in other applications. As requested by the next comment, we also added a case study to this repository, illustrating how to use the software. With those improvements, anyone can download and modify the code to evaluate the impacts of changes in code.

Concerning the parameter alpha, it is actually user-specifiable. The different choices the user can make are: alpha, the definition of X and Y, the observational dataset used, and the climate model ensemble used (CMIP5, CMIP6, HighResMIP, etc). Following the reviewer's comment, we have clarified in the main text that these parameters are user-configurable when introduced in the article.

My final major comment is that I would like to see a specific example to illustrate the package. The author does provide synthetic examples (e.g., Figure 8), but I think it would be more powerful and interesting if the authors recompute an example from the literature, showing that previous studies have flaws that are significant enough to warrant revisiting.

In response to this comment and a similar one from the second reviewer, we have added a case study. This uses the past (2015–2024 mean) anomaly of global mean surface air temperature (GSAT) to constrain the future (2081–2100 mean) anomaly of GSAT. The reference period is 1850–1900.

Section 5 analyses the differences between ClimLoco1.0 and two methods frequently used in the literature. We use this case study to add a section in the appendix (to avoid having a too large main paper) which illustrates these differences, namely in the consideration of observational noise and uncertainty arising from a limited number of climate models.

Moreover, in response to comments from both reviewers, we added two further tests. (i) First, we compared the results when constrained by one observed variable (GSAT 2015–2024 mean), another variable (GSAT 1970–2014 trend) or both of them. This shows the advantages of using a multivariate approach to reduce uncertainty. It also shows that the results depend on the choice of the observed variable. (ii) We also tested whether the distributions were Gaussian. This may be true for the observations (from HadCRUT 5), but not for climate models. This highlights the importance of enabling ClimLoco to take non-Gaussian distributions into account in a future version.

Other than that, my only comments are that there are a few typos or improper tense issues that the copyediting office can likely take care of.

We have done our best to address these issues in the corrected manuscript.

**Reviewer 2:**

The manuscript presents ClimLoco1.0, a statistical model aimed at providing confidence intervals for climate variables using a multivariate linear observational constraint approach. This work is a commendable effort to address the pressing challenge of reducing uncertainty in climate projections. By extending traditional emergent constraint methods to a multivariate framework, the authors offer an innovative tool that leverages multiple observational constraints to enhance the reliability of future climate predictions. The manuscript demonstrates a robust statistical foundation, with detailed derivations of confidence and prediction intervals, making it a valuable contribution to the field.

The strengths of the manuscript include its innovative methodology, relevance to current climate science challenges, and rigorous statistical approach. The use of multivariate regression to constrain climate variables such as equilibrium climate sensitivity (ECS) or gross primary production (GPP) aligns well with the growing need for more sophisticated tools to handle complex climate data. However, the manuscript could benefit from greater clarity in its assumptions, more practical examples, and a thorough discussion of validation and limitations. While the theoretical framework is sound, its accessibility and applicability could be enhanced to broaden its impact within the climate science community. Therefore, I recommend a 'minor revision' of this manuscript and let the authors revise the manuscript accordingly to strengthen its scientific rigor and utility for the meteorology community.

We thank the reviewer for his constructive comments, which really helped to, hopefully, improve the clarity of the manuscript for the climate community and highlight the limitations of this first version as well as observational constraints in general.

**Specific comments:**

1. The authors should clarify assumptions and their implications. The manuscript relies on key assumptions, such as linearity between climate variables and Gaussian error distributions (Page 27-28). These assumptions are critical but may not always hold in climate systems, which often exhibit non-linear dynamics (e.g., feedback loops) or non-Gaussian distributions (e.g., extreme events). The authors should add a dedicated subsection in the methods or discussion section to explicitly list these assumptions and discuss their implications. For instance, address how non-linear relationships or heavy-tailed distributions might affect the confidence intervals and suggest potential extensions (non-linear regression) for future work. This will improve transparency and help readers assess the model's applicability to diverse climate scenarios.

In response to this comment and a similar one from the first reviewer, we added a section called 'Discussion of the assumptions' after section 4. This new section summarises all the assumptions used in ClimLoco1.0 and discusses their implications, proposing future potential improvement. Regarding the Gaussian assumption, we added the following paragraph:

"The assumption that the distributions are Gaussian is clearly recognized as potentially questionable, e.g. in regard to precipitation. If the distribution is not centred, the confidence interval should not be centred either. If the distribution has significant tailed areas, the limits of the confidence interval must be further apart. The greater the difference between the distribution and a Gaussian distribution, the less accurate ClimLoco1.0 will be in estimating

the limits of the confidence interval. However, we did not estimate here whether this impact is significant or negligible. To address this issue in future development of ClimLoco1.0 using non-Gaussian distributions, we recommend employing a bootstrap method to empirically derive a confidence interval. Bootstrapping involves repeatedly resampling the dataset with replacement to create different sub-datasets. Each sub-dataset yields a different observational constraint result. These distributions are then used to compute a confidence interval. However, in this case, no analytical expression of the confidence interval can be derived since this remains an empirical approach."

While a non-linear relationship between climate variables does not invalidate ClimLoco1.0, it will make it less useful. The linear regression model used in ClimLoco1.0 has an error ( $\epsilon$  in the article) that affects the uncertainty of ClimLoco1.0. As described in the article, the weaker the linear relationship (i.e. the poorer the correlation), the greater the error and the lower the uncertainty reduction. In this case, we recommend using a non-linear regression model. A confidence interval can be computed using a bootstrap method, as described in the previous paragraph. However, as previously mentioned, this is empirical and no analytical expression can be derived.

2. The authors need to provide a detailed case study. The manuscript mentions potential applications but lacks a concrete example demonstrating ClimLoco1.0's implementation. Include a section applying the model to a specific climate variable, such as global mean temperature or precipitation, using real observational data. Present the observational constraints, derived confidence intervals, and a comparison with traditional univariate methods. This would illustrate the model's practical utility, making it more compelling and easier for potential readers to replicate or adapt.

In response to this comment and a similar one from the first reviewer, we have added a case study. This uses the past (2015–2024 mean) anomaly of global mean surface air temperature (GSAT) to constrain the future (2081–2100 mean) anomaly of GSAT. The reference period is 1850–1900.

Section 5 analyses the differences between ClimLoco1.0 and two methods frequently used in the literature. We use this case study to add a section in the appendix (to avoid having a too large main paper) which illustrates these differences, namely in the consideration of observational noise and uncertainty arising from a limited number of climate models.

Moreover, in response to comments from both reviewers, we added two further tests. (i) First, we compared the results when constrained by one observed variable (GSAT 2015–2024 mean), another variable (GSAT 1970–2014 trend) or both of them. This shows the advantages of using a multivariate approach to reduce uncertainty. It also shows that the results depend on the choice of the observed variable. (ii) We also tested whether the distributions were Gaussian. This may be true for the observations (from HadCRUT 5), but not for climate models. This highlights the importance of enabling ClimLoco to take non-Gaussian distributions into account in a future version.

3. The authors should expand on model validation. While the manuscript provides theoretical derivations in Page 27-29, it does not discuss how ClimLoco1.0's performance was validated against real-world data or existing methods. It would be better to add a validation section that evaluates the model's performance, potentially using cross-validation or comparison with established emergent constraint approaches, including metrics like coverage probability or uncertainty reduction to quantify improvements. This would strengthen the manuscript's credibility and provide evidence of the model's effectiveness, addressing a key expectation in scientific reviews.

ClimLoco1.0 aims to clarify rigorously the mathematical and statistical foundations of observational constraints. It is not clear to us what to validate in ClimLoco1.0 using real-world data since, as explained in lines 409 to 412, the distributions of models and observations differ in nature. While it can be an interesting perspective, it might necessitate further methodological developments which are, in our opinion, out of the scope of this already long paper.

A comparison between ClimLoco1.0 and existing methods is already made in the article, from a theoretical point of view in Section 5. The differences are also illustrated in the new Section I2 on the real case study. However, it is difficult to really validate ClimLoco1.0 by comparing its performance against other methods since each method quantifies uncertainty differently.

4. The authors should add a discussion to discuss potential biases and uncertainties. The model's reliance on covariance matrices (in Page 26) and observational constraints introduces potential biases, such as errors in covariance estimation or constraint selection, which are not addressed. The authors should include a discussion on sources of uncertainty (e.g., noisy observations in Page 28) and biases, and conduct a sensitivity analysis to show how these factors affect the confidence intervals and suggest mitigation strategies (e.g., robust estimation). Acknowledging and addressing these issues will enhance the model's robustness and guide users in its application. Also, the authors need to add a subsection in the discussion comparing ClimLoco1.0's methodology and performance to existing multivariate emergent constraint approaches. Highlight unique features, such as its handling of noisy observations (page 28), and discuss trade-offs.

In response to this comment, we have included a discussion on the biases and uncertainties considered or introduced by ClimLoco1.0. As these are a direct implication of the assumptions used to construct ClimLoco1.0, we added this discussion to the new section entitled "Discussion of the assumptions".

In the case study, we conducted tests on the sensitivity of ClimLoco1.0 to the factors we know to have the greatest impact: the choice of different observable variables, observational uncertainty (i.e. signal-to-noise ratio) and the number of climate models. We did not test sensitivity to other factors as this would be too extensive for the present paper, which is already long. Furthermore, it can sometimes be difficult to test (e.g. the impact of assuming Gaussian distributions).

Section 5 of the article already compares our approach with two other approaches often used in the literature. One of these (Ribes et al., 2021) is multivariate. The other multivariate approaches in the literature are mainly based on nonlinear methods and are not compared to our method due to their empirical approaches which make it difficult to compare them. We consequently added a paragraph to section 5 of the article discussing the differences between our approach and the methodology of these non-linear multivariate approaches.

**5. Enhance accessibility for broader readers**. The manuscript's statistical derivations (Page 26-29) are rigorous but may be inaccessible to climate scientists if someone without a strong statistical background. The authors need to include a brief primer or appendix explaining key concepts (e.g., multivariate regression, confidence vs. prediction intervals) in simpler terms. In addition, using visual aids, such as a flowchart of the ClimLoco1.0 workflow, to complement the equations. This will broaden the impact of this study, making the model more usable by meteorologists and policymakers who may not be statisticians.

We identified all the key statistical concepts necessary for understanding how ClimLoco1.0 is built, including realisations of a random variable, estimators, probability distributions, conditional probability distributions, quantiles, probability intervals, confidence intervals, Gaussian distributions, Student's t-distributions, and finally, multivariate linear regression. We added an appendix with a figure to illustrate these concepts.

Furthermore, we included a figure describing the flowchart of the article in the introduction.

---

## Author Response (AR2)

As required by the editor, the figures 8, 9 and 10 have been modified, as well as their caption, to ensure that readers with colour vision deficiencies can correctly interpret our findings. The widths and lengths of the lines has been modified.